# Selective neuronal degeneration in MATR3 S85C knock-in mouse model of early-stage ALS

Ching Serena Kao[1,2,4], Rebekah van Bruggen[1,4], Jihye Rachel Kim[1,2,4], Xiao Xiao Lily Chen [1,2,4], Cadia Chan [1,2], Jooyun Lee [1], Woo In Cho [1], Melody Zhao[1,2], Claudia Arndt[1], Katarina Maksimovic[1,2], Mashiat Khan[1,2], Qiumin Tan[3], Michael D. Wilson [1,2] & Jeehye Park [1,2✉]

A missense mutation, S85C, in the *MATR3* gene is a genetic cause for amyotrophic lateral sclerosis (ALS). It is unclear how the S85C mutation affects MATR3 function and contributes to disease. Here, we develop a mouse model that harbors the S85C mutation in the endogenous *Matr3* locus using the CRISPR/Cas9 system. MATR3 S85C knock-in mice recapitulate behavioral and neuropathological features of early-stage ALS including motor impairment, muscle atrophy, neuromuscular junction defects, Purkinje cell degeneration and neuroinflammation in the cerebellum and spinal cord. Our neuropathology data reveals a loss of MATR3 S85C protein in the cell bodies of Purkinje cells and motor neurons, suggesting that a decrease in functional MATR3 levels or loss of MATR3 function contributes to neuronal defects. Our findings demonstrate that the MATR3 S85C mouse model mimics aspects of early-stage ALS and would be a promising tool for future basic and preclinical research.

[1] Genetics and Genome Biology Program, The Hospital for Sick Children, Toronto, ON, Canada. [2] Department of Molecular Genetics, University of Toronto, Toronto, ON, Canada. [3] Department of Cell Biology, University of Alberta, Edmonton, AB, Canada. [4]These authors contributed equally: Ching Serena Kao, Rebekah van Bruggen, Jihye Rachel Kim, Xiao Xiao Lily Chen. ✉email: jeehye.park@sickkids.ca

 

A key feature of neurodegenerative diseases is selective neuronal vulnerability[1,2]. Many disease-causing proteins have been identified and studied to understand how these proteins render certain neurons more vulnerable than others and lead to progressive defects in specific brain functions, despite their ubiquitous expression throughout the nervous system. Amyotrophic lateral sclerosis (ALS) is one such disease, featuring selective vulnerability of motor neurons accompanied by progressive motor deficits, paralysis, respiratory failure and death 3–5 years after disease onset[3]. Evidence from both ALS patients and animal models suggests that degeneration of motor neurons begins at the neuromuscular junctions (NMJs) within the muscle, then proceeds through the axons towards the cell bodies of the motor neurons, advocating ALS as a disease of "dying-back" motor neuronopathy[4–8]. However, why motor neurons are selectively vulnerable in ALS and how degeneration occurs is not clearly understood.

To date, about 30 ALS-linked genes have been identified, including superoxide dismutase 1 (SOD1), TAR DNA-binding protein 43 (TDP-43), fused in sarcoma (FUS) and chromosome 9 open reading frame 72 (C9orf72)[9,10]. Genetic discoveries have allowed for the generation of disease models for understanding ALS pathogenesis. SOD1 G93A mice were the first transgenic model of ALS. They recapitulate many aspects of the disease and have been widely used for disease studies and preclinical research[11,12]. Many other transgenic models (e.g., TDP-43 and FUS) have provided further insights into the disease process but showed limitations[13–16]. Transgenic ALS models usually contain multiple copies of exogenous disease-linked genes and express high levels of proteins under a constitutively active promoter. Overexpression of wild-type and mutant proteins show similar degrees of phenotypic severity, suggesting that toxicity arises mostly from overexpression rather than from the mutation itself. Therefore, unraveling the precise role of disease-causing proteins and the pathomechanisms of progressive motor neuron degeneration has been challenging using transgenic animal models.

MATR3 has been recently linked to ALS[17]. It encodes a component of the nuclear matrix and contains two RNA recognition motifs and two zinc finger domains[18,19]. Recent studies have highlighted the role of MATR3 in regulating alternative splicing[20,21]. About a dozen missense mutations in MATR3 have been identified in familial and sporadic ALS patients[22–28]. The S85C mutation is the most frequently identified ALS-linked mutation in MATR3. It was originally identified in several cases of familial distal myopathy that were later reclassified as familial ALS[22,29–31]. The S85C mutation is inherited in an autosomal dominant manner[29–31]. Patients with the S85C mutation have an earlier age of onset (35–50 years) compared to those carrying other ALS-linked mutations[29,31]. An F115C mutation in MATR3 is associated with familial ALS and dementia[22]. However, little is known about the pathogenic potential of these missense mutations in MATR3 and the mechanisms by which these mutations affect MATR3 function and cause disease.

A few studies have investigated whether ALS-linked mutations in MATR3 confer pathogenicity, and if so, how these mutations cause disease. In rat primary cortical neurons, expression of ALS-linked mutant MATR3 (e.g., S85C and F115C) was found to be more toxic than wild-type MATR3[32], suggesting that the mutations are pathogenic. This study also showed that although the ALS-linked mutations do not alter MATR3 localization, the S85C mutation reduces MATR3 solubility[32]. The pathogenic potential of ALS-linked mutations in MATR3 was also explored in mice. Transgenic mice that overexpress human MATR3 F115C in muscles exhibited muscle atrophy associated with vacuoles and internalized nuclei; however, mice that overexpress human wild-type MATR3 displayed a similar phenotype albeit to a lesser

degree[33]. Mice that were injected with adeno-associated viruses overexpressing human MATR3 S85C specifically in the muscle displayed myogenic changes such as smaller myofibers with internalized nuclei, similar to mice overexpressing wild-type MATR3[34]. These studies suggest that the phenotypic changes largely result from overexpression of MATR3 and not from the ALS-linked mutation. Therefore, to better delineate the pathogenic nature of the S85C mutation in MATR3, there is a need for a model that expresses mutant MATR3 at the physiological level.

Here we generated a MATR3 S85C germline knock-in mouse line by introducing the S85C mutation in the endogenous mouse Matr3 locus using the CRISPR/Cas9 system. This enables the expression of MATR3 S85C under its endogenous promoter, thereby maintaining its expression at the physiological level. Therefore, we created a MATR3 S85C knock-in model, which closely resembles the human ALS genotype. We conducted extensive behavioral and neuropathological studies on our S85C knock-in mice to determine the key events in disease progression and to investigate the role of MATR3 S85C in disease pathogenesis. Homozygous S85C mice exhibited motor deficits with hind-limb dragging and NMJ defects, reminiscent of early pathogenic events in ALS. The mutant mice also showed a striking loss of Purkinje cells in the cerebellum. Intriguingly, loss of MATR3 S85C was observed specifically in Purkinje cells and motor neurons in these mutant animals. These results suggest that mutant MATR3 is not tolerated specifically in Purkinje cells and motor neurons, and that the S85C mutation causes disease through loss of MATR3 function.

## Results

**MATR3 is expressed in the mouse nervous system**. We investigated the expression pattern of MATR3 in the brain and spinal cord of wild-type C57BL/6J mice. First, several commercially available MATR3 antibodies were tested on wild-type and Matr3 knockout mice to determine antibody specificity. Matr3 knockout mice were generated on the C57BL/6J genetic background using the CRISPR/Cas9 system, whereby an insertion or deletion within the Matr3 coding sequence resulted in a frameshift and introduced a premature stop codon. Three founders (p.L122Wfs*5, 10 bp insertion; p.F115Lfs*7, 4 bp deletion; p.F87Lfs*34, 7 bp insertion) were germline transmitted and backcrossed four times to wild-type C57BL/6J mice to remove any off-target effects. Timed mating experiments determined that Matr3 knockout mice of all three alleles were perinatal lethal, demonstrating the requirement for MATR3 expression during embryonic development (Supplementary Fig. 1). We verified the specificity of two MATR3 antibodies through western blotting analysis and immunohistochemistry: HPA036565 that recognizes amino acids 145–233 and ab84422 that recognizes 800–847, hereafter referred to as MATR3-N and MATR3-C antibodies, respectively (Supplementary Fig. 2). MATR3 is ubiquitously expressed in wild-type embryos with strong expression in the central nervous system (Supplementary Fig. 2b, c). We next examined the expression of MATR3 in postnatal 6-week-old brain and spinal cord of wild-type mice. MATR3 is ubiquitously expressed throughout the brain including the cortex, hippocampus and cerebellum, as well as the spinal cord (Supplementary Figs. 3a and 4), and predominantly localized in the nucleus of cells in the nervous system, similar to previously reported results (Supplementary Figs. 3a and 4)[35].

**Generation of MATR3 S85C knock-in mice**. To investigate the consequences of the S85C mutation in MATR3, we introduced the mutation into the mouse Matr3 endogenous locus using CRISPR/Cas9. Mouse MATR3 protein is ~98% identical to human MATR3 and, importantly, S85 and the surrounding

amino acids are highly conserved (Fig. 1a). MATR3 S85C knock-in mice have a c.254C > G change that introduces the desired amino acid change (Fig. 1b, c). These mice also have two silent mutations (c.261T > C, p.F87 and c.264C > T, p.N88), which prevent re-cleavage of the repaired allele and simultaneously remove an MseI restriction endonuclease site, allowing for gen-otyping (Supplementary Fig. 5). A single founder male was found to harbor the S85C mutation in the germline and was backcrossed four times to wild-type C57BL/6J mice to remove any off-target effects. Potential off-target sites of the guide RNA, identified by computational tools, were confirmed to be unmodified through sequencing (Supplementary Data 1). In addition, we verified that all *Matr3* exons are intact through sequencing (Supplementary Data 2).

At the fifth generation, MATR3 S85C heterozygous male and female mice were intercrossed to obtain S85C homozygous and heterozygous mice, as well as wild-type littermates (genotypes are *Matr3*$^{S85C/S85C}$, *Matr3*$^{S85C/+}$ and *Matr3*$^{+/+}$, respectively). *Matr3*$^{S85C/S85C}$ mice allow us to assess the effect of the S85C mutation in the absence of the wild-type allele. Offspring were born at the expected genotype frequencies, indicating that the S85C mutation does not cause perinatal lethality like the null allele does (Fig. 1d). As the S85C knock-in mouse retains all the regulatory regions of the mouse *Matr3* gene including its endogenous promoter and 3′-untranslated region, we assessed *Matr3* mRNA levels and MATR3 protein levels in the bulk brain and spinal cord tissue at 3 weeks of age. Both mRNA and protein levels of MATR3 were similar between the three genotypes (Fig. 1e, f and Supplementary Fig. 6), indicating that the S85C mutation does not affect endogenous levels of MATR3 in either *Matr3*$^{S85C/+}$ or *Matr3*$^{S85C/S85C}$ mice.

**Matr3$^{S85C/S85C}$ mice reach end-stage earlier**. Next, we investi-gated whether MATR3 S85C knock-in mice recapitulate any ALS features including weight loss and health decline[26]. As ALS patients experience weight loss during disease progression, we monitored the weight of animals starting at 3 weeks of age and continued bi-weekly. During the first 3–4 months, *Matr3*$^{S85C/S85C}$ animals appeared normal and underwent rapid weight gain like wild-type mice. However, size differences were apparent from 20 weeks of age and onwards (Fig. 2). *Matr3*$^{S85C/S85C}$ mice failed to gain weight at the same rate as *Matr3*$^{S85C/+}$ and *Matr3*$^{+/+}$ mice starting at 15 and 21 weeks of age for males and females, respectively (Fig. 2a). The weight of *Matr3*$^{S85C/S85C}$ animals plateaued until 60 weeks of age, at which point some of the mice began to significantly lose weight and reached endpoint (Fig. 2). Mice were scored weekly for body condition, weight loss, activity/mobility and appearance: a combined score of 60 was defined as the endpoint. *Matr3*$^{S85C/S85C}$ animals showed scores averaging around 40 by 60 weeks of age (Fig. 2b and Supplementary Data 3).

**Matr3$^{S85C/S85C}$ mice exhibit progressive motor deficits**. We conducted various motor behavioral tests at different ages to determine the time of disease onset and the stages of disease progression. We tested motor coordination and balance using the Rotarod assay that measures the length of time before a mouse falls from an accelerating rod (at 6, 10, 20 and 30 weeks of age). We also conducted the Dowel test that assesses motor coordi-nation by measuring the number of times a mouse crosses a dowel in 2 min (at 10, 20, and 30 weeks of age). Starting at 10 weeks of age, *Matr3*$^{S85C/S85C}$ mice exhibited a significantly reduced latency to fall compared to *Matr3*$^{S85C/+}$ and *Matr3*$^{+/+}$ mice, and progressively worsened with age (Fig. 3a and

Supplementary Fig. 7a). Correspondingly, *Matr3*$^{S85C/S85C}$ animals were significantly slower and had greater difficulty crossing the dowel and progressively worsened over time (Fig. 3b and Sup-plementary Fig. 7b). In addition, we conducted an open-field assay to assess rearing activity. *Matr3*$^{S85C/S85C}$ male and female mice exhibited significantly reduced length of time standing on their hind limbs starting at 10 weeks of age. The vertical time decreased with age, suggesting hind-limb weakness in *Matr3*$^{S85C/S85C}$ mice (Fig. 3c and Supplementary Fig. 7c).

At 20 weeks of age, ~73% of all *Matr3*$^{S85C/S85C}$ mice began to show a limb clasping phenotype (Supplementary Fig. 7d–f). Furthermore, *Matr3*$^{S85C/S85C}$ mice showed a significant reduction in muscle strength at 30 and 55 weeks of age using the inverted grid test (Fig. 3d, e). A delayed righting reflex was observed at 40 weeks of age in *Matr3*$^{S85C/S85C}$ animals, whereby animals were flipped onto their side or back and showed delayed (>1 s) righting. The righting time increased with age and was observed in almost all *Matr3*$^{S85C/S85C}$ animals by 60 weeks of age (Fig. 3f). In addition to difficulty standing on their hind limbs, reduced muscle strength and delayed righting, *Matr3*$^{S85C/S85C}$ mice exhibited ataxia, severe gait disturbances and occasional hind-limb dragging near end-stage (Fig. 3g–i and Supplementary Movies 1 and 2). Together, these results suggest that *Matr3*$^{S85C/S85C}$ mice reflect progressive motor impairments observed in ALS patients.

Given that ALS patients may also develop dementia[22,36], we conducted a memory test on *Matr3*$^{S85C/S85C}$ mice at 40 weeks of age. The contextual and cued fear conditioning test is a classical paradigm that assesses associative fear learning and memory in mice[37,38]. Male *Matr3*$^{S85C/S85C}$ animals showed significantly reduced freezing time in both the contextual and cued tests compared to wild-type mice, demonstrating memory deficits (Supplementary Fig. 7g). However, female *Matr3*$^{S85C/S85C}$ animals exhibited no differences in both contextual and cued tests. This result is unexpected as the S85C mutation has not been associated with male-specific dementia. Further studies are warranted to validate and examine memory deficits in these mutant mice.

**Motor-related neuronal defects in Matr3$^{S85C/S85C}$ mice**. Given the impaired motor function observed in aged *Matr3*$^{S85C/S85C}$ mice, we investigated whether *Matr3*$^{S85C/S85C}$ mice show neuro-pathological changes in the neurons that govern motor function in the brain and spinal cord. Interestingly, immunostaining results from 60-week-old whole brains showed a significant reduction in the size of the cerebellum of *Matr3*$^{S85C/S85C}$ mice compared to *Matr3*$^{S85C/+}$ and *Matr3*$^{+/+}$ mice (Fig. 4d, e and Supplementary Fig. 3b), but not at 6 weeks of age when *Matr3*$^{S85C/S85C}$ mice do not show motor phenotypes (Fig. 4a, b and Supplementary Fig. 3a). As Purkinje cells of the cerebellum play a key role in regulating and coordinating movement and have been found to be affected in ALS[39–42], we wanted to see if Purkinje cells were degenerated in *Matr3*$^{S85C/S85C}$ mice. Intriguingly, immunostaining of the cere-bellum with anti-calbindin antibody showed an extensive loss of Purkinje cells in *Matr3*$^{S85C/S85C}$ mice (Fig. 4d, f), whereas no sig-nificant change was observed at 6 weeks of age (Fig. 4a, c). Moreover, staining of neurofilaments and myelin revealed axonal spheroids in the granule cell layer of *Matr3*$^{S85C/S85C}$ mice at disease end-stage, which may indicate compromised axonal integrity in the Purkinje cells (Supplementary Fig. 8).

The major pathological feature of ALS is motor neuron degeneration. We examined if there was loss of motor neuron cell bodies in the ventral region of the lower lumbar spinal cord (L4–L6) in *Matr3*$^{S85C/S85C}$ mice at disease end-stage. Among several different types of motor neurons, α-motor neurons are

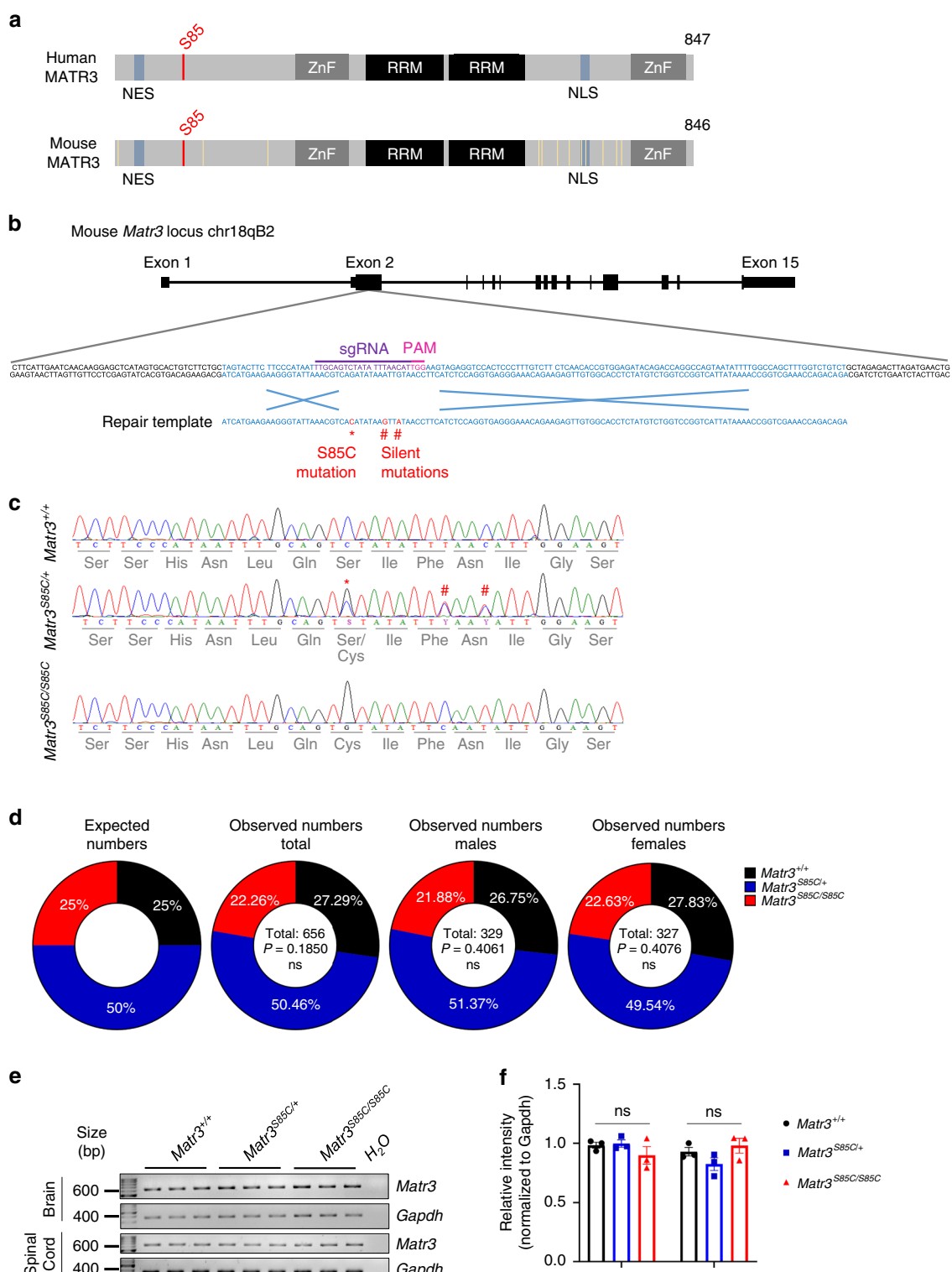

**Fig. 1 Generation of MATR3 S85C knock-in mice using CRISPR-Cas9 technology. a** Schematic representation comparing the identity of protein sequences between human and mouse MATR3. The serine 85 residue in red is conserved between humans and mice. Yellow lines in mouse MATR3 indicate residues that are not conserved in human MATR3. Nuclear export sequence, NES; zinc finger motif, ZnF; RNA recognition motif, RRM; nuclear localization sequence, NLS. **b** A guide RNA and repair template were designed to introduce the S85C mutation (red asterisk) within the mouse *Matr3* locus (Chr18qB2) in exon 2. The addition of two silent mutations (red hash) disrupted the PAM sequence and prevented re-cleavage of the repaired template. **c** Sanger sequence verification of the S85C and silent mutations in the endogenous mouse *Matr3* locus. **d** *Matr3*^S85C/S85C mice are born at the expected Mendelian ratio. No significant (ns) differences between observed and expected numbers using the chi-squared test for goodness of fit. **e** *Matr3* mRNA levels in the brain and spinal cord at 3 weeks of age by RT-PCR. *Gapdh* primers were used as a control. **f** Quantification of **e**, $n = 3$ biological replicates per genotype. Data are presented as mean ± SEM and show no significant (ns) differences in *Matr3* mRNA levels between the littermates as determined by unpaired two-tailed *t*-test. Source data are provided as a Source Data file.

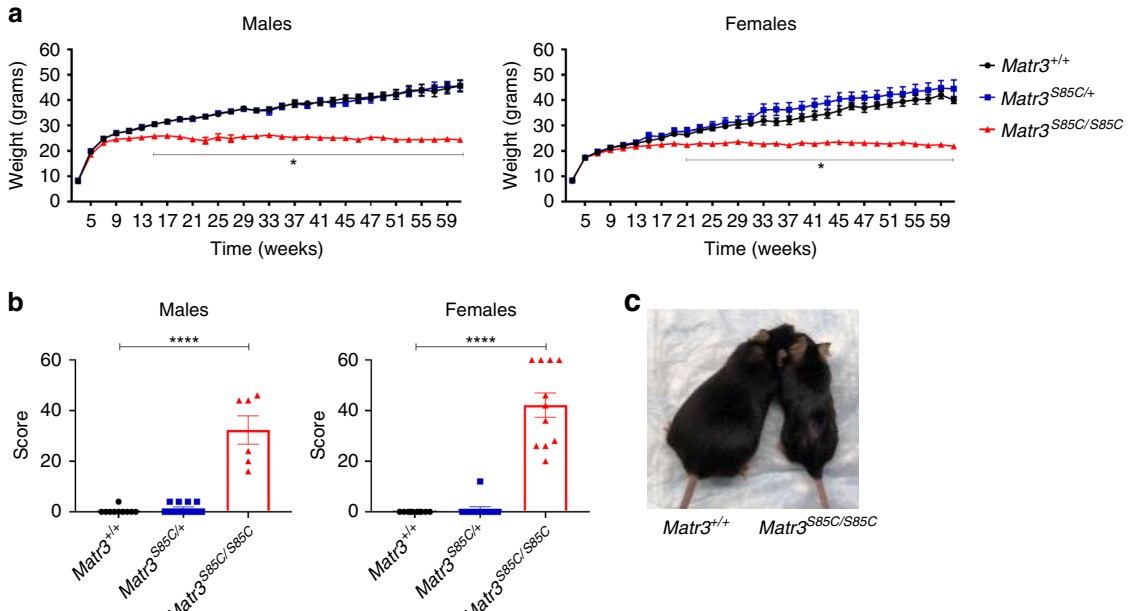

**Fig. 2 Homozygous S85C knock-in mice reach humane endpoint at over one year of age. a** Homozygous S85C animals (both males and females) exhibited significant differences in weight starting at around 15 and 21 weeks of age for males and females, respectively. Each dot represents mean ± SEM, where males: $n$ = at least 10 $Matr3^{+/+}$, 10 $Matr3^{S85C/+}$, 5 $Matr3^{S85C/S85C}$; females: $n$ = at least 8 $Matr3^{+/+}$, 10 $Matr3^{S85C/+}$, 8 $Matr3^{S85C/S85C}$ (exact $n$ numbers for each age are presented in Supplementary Information). Two-way ANOVA, Dunnett correction for multiple comparisons, see Supplementary Information for $p$-values at each age (*$p$ < 0.05). **b** The phenotype score for wild-type, heterozygous and homozygous S85C male and female animals at over 1 year of age. Mice with a score of 60 were considered to be at the endpoint. Each dot represents a single animal (males: $n$ = 10 $Matr3^{+/+}$, 12 $Matr3^{S85C/+}$, 6 $Matr3^{S85C/S85C}$; females: $n$ = 10 $Matr3^{+/+}$, 12 $Matr3^{S85C/+}$, 11 $Matr3^{S85C/S85C}$), with data presented as mean ± SEM. Significance was determined by unpaired two-tailed $t$-test; ****$p$ < 0.0001. **c** Size difference between wild-type and homozygous S85C mice at endpoint. Source data are provided as a Source Data file.

large neurons that innervate fast-twitching muscle fibers and are the most vulnerable neurons in ALS patients, whereas γ-motor neurons are smaller neurons that innervate muscle spindles and are spared in ALS[43]. Both α- and γ-motor neurons are marked by ChAT, but α-motor neurons are molecularly distinguished by higher NeuN expression relative to γ-motor neurons[43,44]. There was a trend towards decrease in the number of α-motor neurons, but not γ-motor neurons, in $Matr3^{S85C/S85C}$ mice when compared to wild-type mice at end-stage (Fig. 5a, b). We also assessed NMJs of the tibialis anterior (TA) muscle, which comprised primarily of fast-twitch muscle fibers innervated by α-motor neurons of the lower lumbar spinal cord and are found to be more susceptible to disease[1,4,45]. Intriguingly, $Matr3^{S85C/S85C}$ mice showed a significant increase in the percentage of partially denervated endplates compared to wild-type littermates (Fig. 5c, d and Supplementary Fig. 9), indicating that NMJs are affected in $Matr3^{S85C/S85C}$ mice. In addition, we found a significant increase in the number of neurofilament accumulations near the presynaptic terminals in $Matr3^{S85C/S85C}$ mice, suggesting axonal degeneration in $Matr3^{S85C/S85C}$ mice (Fig. 5e, f). Immunostaining with α-bungarotoxin (BTX) revealed that $Matr3^{S85C/S85C}$ mice show significantly smaller endplate sizes in TA muscles compared to wild-type mice (Fig. 5g, h). Altogether, these results suggest that Purkinje cells and α-motor neurons are compromised in $Matr3^{S85C/S85C}$ mice.

**MATR3 S85C loss precedes neuronal loss in $Matr3^{S85C/S85C}$ mice.** Next, we wanted to investigate whether the S85C mutation alters MATR3 levels or localization in the affected neurons of both the cerebellum and spinal cord. First, we co-immunostained with anti-calbindin and anti-MATR3 antibodies to examine MATR3 S85C localization in Purkinje cells at disease end-stage.

Wild-type MATR3 is predominantly localized to the nucleus in the brain and spinal cord of $Matr3^{+/+}$ mice (Fig. 6a). Interestingly, although there was no change in MATR3 S85C localization in the cells of the granular and molecular layer of the cerebellum, there was a striking loss of MATR3 staining in the nucleus of Purkinje cells in $Matr3^{S85C/S85C}$ mice (Fig. 6a, c). To determine whether loss of MATR3 staining precedes Purkinje cell loss in $Matr3^{S85C/S85C}$ mice, we performed immunostaining on 6-week-old brains. Although there was no significant Purkinje cell loss observed at this stage (Fig. 4a–c), drastically reduced staining of MATR3 S85C in Purkinje cells was observed (Fig. 6b, d and Supplementary Fig. 3a). This was validated using both anti-MATR3-N and anti-MATR3-C antibodies (Fig. 6b, Supplementary Fig. 10 and Supplementary Data 4). In addition, we found reduced MATR3 staining in most (~80%) Purkinje cells throughout all lobules of the cerebellum in $Matr3^{S85C/S85C}$ mice at 6 weeks of age (Fig. 6d and Supplementary Fig. 10a) and this finding was more obvious at older ages (Supplementary Fig. 11). Taken together, these results suggest that the S85C mutation leads to loss of MATR3 protein, which in turn contributes to progressive degeneration of Purkinje cells in $Matr3^{S85C/S85C}$ mice.

**MATR3 pathology in α-motor neurons in $Matr3^{S85C/S85C}$ mice.** In addition to Purkinje cell loss, $Matr3^{S85C/S85C}$ mice show motor neuron defects (Fig. 5). Therefore, we wanted to investigate whether MATR3 pathology is observed in α-motor neurons but not in γ-motor neurons in $Matr3^{S85C/S85C}$ mice. Indeed, immunostaining results showed significantly reduced MATR3 staining in the ChAT- and NeuN-positive α-motor neurons, but not in the NeuN-negative γ-motor neurons, within the lumbar spinal cord of $Matr3^{S85C/S85C}$ mice at disease end-stage (Fig. 6e–g). These

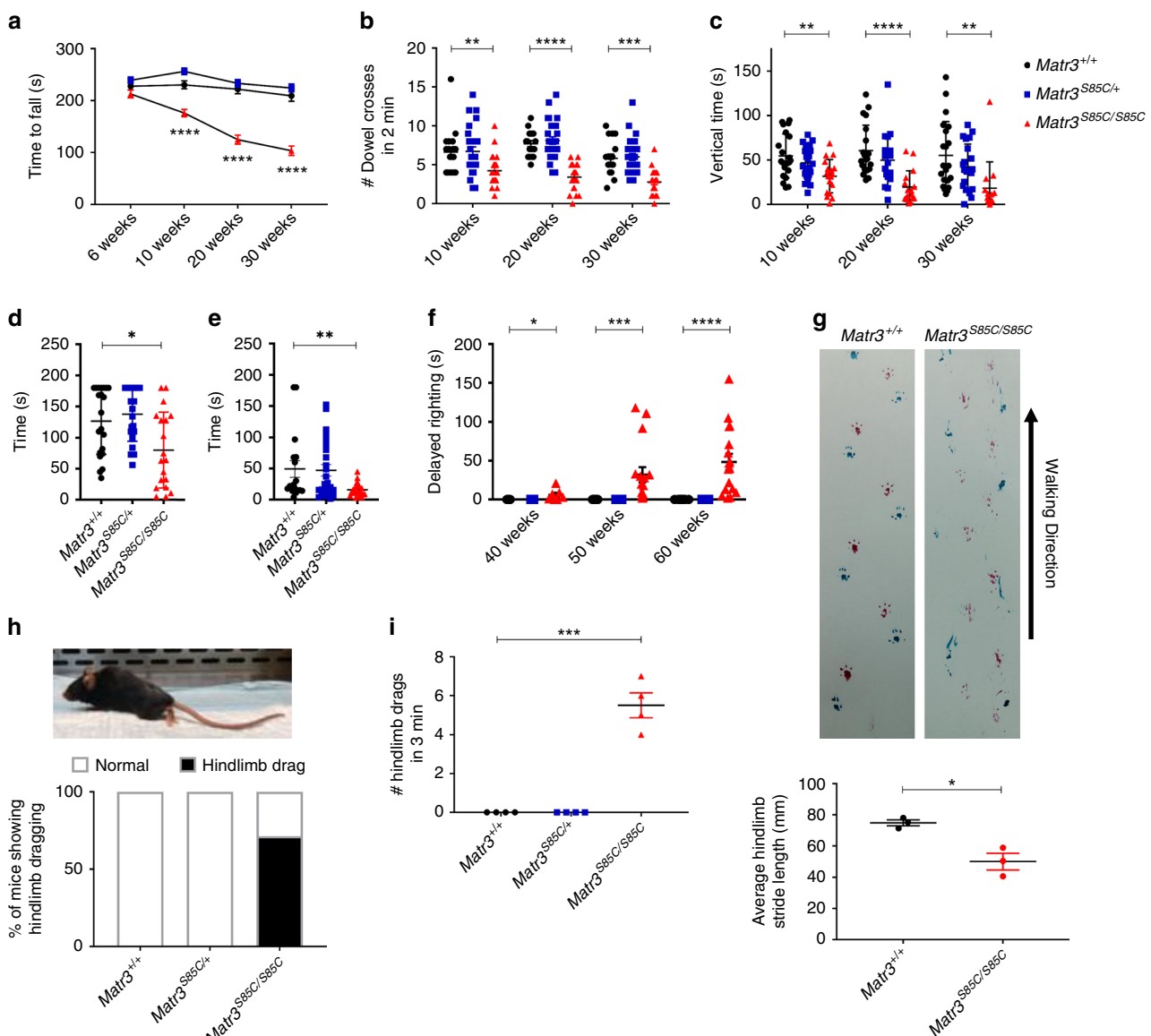

**Fig. 3 Homozygous S85C knock-in mice exhibit progressive motor deficits. a**, **b** Motor coordination was assessed by (**a**) Rotarod (6 weeks: $n = 20$ Matr3$^{+/+}$, 30 Matr3$^{S85C/+}$, 24 Matr3$^{S85C/S85C}$; 10 weeks: $n = 19$ Matr3$^{+/+}$, 29 Matr3$^{S85C/+}$, 23 Matr3$^{S85C/S85C}$, ****$p < 0.0001$; 20 weeks: $n = 17$ Matr3$^{+/+}$, 27 Matr3$^{S85C/+}$, 22 Matr3$^{S85C/S85C}$, ****$p < 0.0001$; 30 weeks: 17 Matr3$^{+/+}$, 27 Matr3$^{S85C/+}$, 22 Matr3$^{S85C/S85C}$, ****$p < 0.0001$) and by (**b**) dowel test (10 weeks: $n = 19$ Matr3$^{+/+}$, 28 Matr3$^{S85C/+}$, 23 Matr3$^{S85C/S85C}$, **$p = 0.0057$; 20 weeks: $n = 17$ Matr3$^{+/+}$, 27 Matr3$^{S85C/+}$, 22 Matr3$^{S85C/S85C}$ ****$p < 0.0001$; 30 weeks: 17 Matr3$^{+/+}$, 27 Matr3$^{S85C/+}$, 22 Matr3$^{S85C/S85C}$, ***$p = 0.0001$). **c** Rearing time (time standing on hind limbs) was measured using Open-field analysis (10 weeks: 21 Matr3$^{+/+}$, 22 Matr3$^{S85C/+}$, 18 Matr3$^{S85C/S85C}$, **$p = 0.0027$; 20 weeks: 20 Matr3$^{+/+}$, 19 Matr3$^{S85C/+}$, 16 Matr3$^{S85C/S85C}$, ****$p < 0.0001$; 30 weeks: 21 Matr3$^{+/+}$, 22 Matr3$^{S85C/+}$, 14 Matr3$^{S85C/S85C}$, **$p = 0.0044$). **d**, **e** Muscle strength was tested using the inverted grid test at **d** 30 weeks of age ($n = 21$ Matr3$^{+/+}$, 23 Matr3$^{S85C/+}$, 15 Matr3$^{S85C/S85C}$, *$p = 0.0146$) and **e** 55 weeks of age ($n = 17$ Matr3$^{+/+}$, 25 Matr3$^{S85C/+}$, 22 Matr3$^{S85C/S85C}$, **$p = 0.0088$). **f** Delayed righting reflex was measured as the time an animal takes to right itself after being flipped on each side (40 weeks: $n = 13$ Matr3$^{+/+}$, 11 Matr3$^{S85C/+}$, 7 Matr3$^{S85C/S85C}$, *$p = 0.0102$; 50 weeks: $n = 21$ Matr3$^{+/+}$, 23 Matr3$^{S85C/+}$, 16 Matr3$^{S85C/S85C}$, ***$p = 0.0006$; 60 weeks: 20 Matr3$^{+/+}$, 25 Matr3$^{S85C/+}$, 17 Matr3$^{S85C/S85C}$, ****$p < 0.0001$). **g** Representative image of the footprint analysis for homozygous S85C and wild-type mice. The hind-limb stride length (mm) was measured for five consecutive steps, for two replicates per mouse and three mice per genotype. **h** Example of a homozygous S85C knock-in mouse dragging its hind limb. Graph shows percentage of end-stage animals exhibiting hind-limb dragging (n = 7 Matr3$^{+/+}$, 8 Matr3$^{S85C/+}$, 7 Matr3$^{S85C/S85C}$, *$p = 0.0113$). **i** Graph shows the number of times the mice drag their hind limbs in a 3 min time span ($n = 4$ Matr3$^{+/+}$, 4 Matr3$^{S85C/+}$, 4 Matr3$^{S85C/S85C}$, ***$p = 0.0001$). Data shown in **a** to **i** are displayed as mean ± SEM. A single dot represents a single animal in **b**–**i**. Significance was determined using an unpaired two-tailed $t$-test. Source data are provided as a Source Data file.

results suggest that loss of MATR3 function contributes to NMJ defects in TA muscles in Matr3$^{S85C/S85C}$ mice. In addition, we found MATR3 S85C aggregation in the nucleus of a subset of α-motor neurons (Fig. 6h, i), suggesting that gain of MATR3 function may also contribute to axon cytoskeleton defects and denervation of the α-motor neurons in Matr3$^{S85C/S85C}$ mice.

**Muscle atrophy in Matr3$^{S85C/S85C}$ mice.** The S85C mutation has been linked to both ALS and distal myopathy[22,30,31,46]. Therefore, we examined the TA and gastrocnemius (GA) muscles of end-stage mice through hematoxylin and eosin (H&E) staining. We found that the average TA and GA muscle fiber sizes were significantly reduced in Matr3$^{S85C/S85C}$ mice

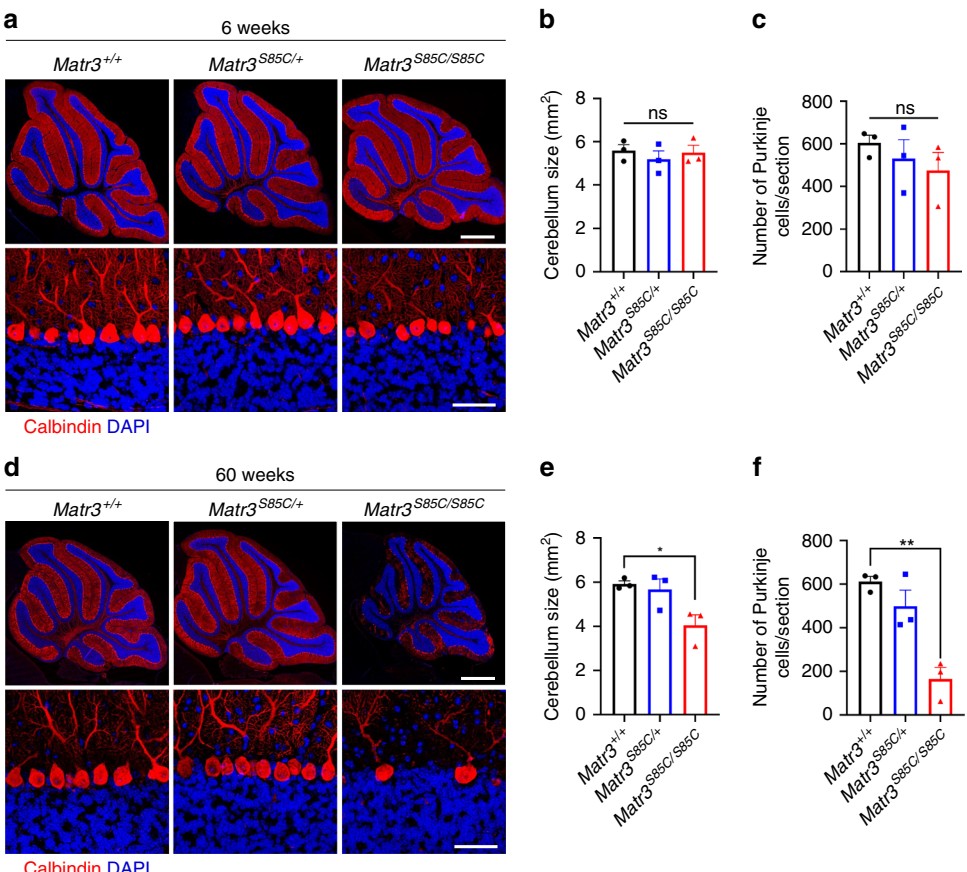

**Fig. 4 A striking loss of Purkinje cells in homozygous S85C mice. a** Representative images of the whole cerebellum (upper row) and magnified images of calbindin-positive Purkinje cells (bottom row) of 6-week-old mice. Scale bars indicate 800 μm (upper row) and 50 μm (bottom row). **b, c** Graphs show (**b**) cerebellum size ($n = 3$ Matr3[+/+], 3 Matr3[S85C/+], 3 Matr3[S85C/S85C], ns = not significant) and (**c**) the total number of calbindin-positive Purkinje cells in the whole cerebellum ($n = 3$ Matr3[+/+], 3 Matr3[S85C/+], 3 Matr3[S85C/S85C], ns = not significant) of 6-week-old mice. **d** Representative images of the whole cerebellum (upper row) and magnified images of calbindin-positive Purkinje cells (bottom row) of endpoint mice (or over 60 weeks of age). Scale bars indicate 800 μm (upper row) and 50 μm (bottom row). **e, f** Graphs show (**e**) cerebellum size ($n = 3$ Matr3[+/+], 3 Matr3[S85C/+], 3 Matr3[S85C/S85C], *$p = 0.0194$) and (**f**) the total number of calbindin-positive Purkinje cells in the whole cerebellum ($n = 3$ Matr3[+/+], 3 Matr3[S85C/+], 3 Matr3[S85C/S85C], **$p = 0.0015$) of endpoint mice. Data shown in **b**, **c**, **e** and **f** are displayed as mean ± SEM and each dot represents a single animal. Significance was determined by unpaired two-tailed *t*-test. Source data are provided as a Source Data file.

(Supplementary Fig. 12b, c, e). However, no prominent myopathic changes were observed, including internalized nuclei and rimmed vacuoles (Supplementary Fig. 12b, d, f). We also examined MATR3 localization and staining intensity in the TA muscles. MATR3 S85C was predominantly localized to the nuclei in the muscles of Matr3[S85C/S85C] mice and its staining intensity was not affected (Supplementary Fig. 12g). Although muscle atrophy in Matr3[S85C/S85C] mice is reminiscent of mild cases of MATR3 S85C-linked myopathy[46], the lack of prominent myopathic features and unaltered levels of MATR3 S85C suggest that muscle atrophy may be secondary to pathological changes in other tissues or cell types, such as the motor neurons.

**Inflammatory genes are upregulated in Matr3[S85C/S85C] mice.** To investigate the molecular changes that may contribute to the behavioral deficits and neuronal defects observed in Matr3[S85C/S85C] mice, we performed RNA sequencing (RNA-seq) on the brain cortex, cerebellum and spinal cord tissue ($n = 4$ for each genotype) at the early symptomatic stage (8–10 weeks of age)[47]. Using a cutoff of fold change>1.5 and false discovery rate (FDR) < 0.05, 110 and 14 differentially expressed genes (DEGs) were

found in the cerebellum and spinal cord, respectively, when comparing Matr3[+/+] and Matr3[S85C/S85C] mice (Fig. 7a, b and Supplementary Data 5). DEGs in both the cerebellum and spinal cord of Matr3[S85C/S85C] mice enriched for Gene Ontology (GO) terms related in immune system pathways (Fig. 7d, e). Several immune response genes (e.g., Mpeg1, a macrophage-specific marker[48], and Clec7a, involved in the innate immune response[49]) were upregulated in both the cerebellum and spinal cord of Matr3[S85C/S85C] mice compared to Matr3[+/+] mice (Fig. 7a, b and Supplementary Data 5), suggesting that neuroinflammation could contribute to disease pathogenesis. Genes that are involved in transmembrane transport were downregulated in the spinal cord (Fig. 7b, e), while we found no enrichment of downregulated genes in the cerebellum (Fig. 7a, d).

In addition, two and five DEGs were found in the cerebellum and spinal cord, respectively, between Matr3[+/+] and Matr3[S85C/+] mice (Supplementary Fig. 13b, d). This finding was consistent with the lack of phenotypic and pathological differences between the two genotypes at 8–10 weeks of age. The top upregulated and downregulated genes observed in Matr3[S85C/S85C] mice compared to Matr3[S85C/+] mice were similar to those found between Matr3[S85C/S85C] and Matr3[+/+] mice, which was also expected based on their phenotypes (Fig. 7 and Supplementary Fig. 13a, c).

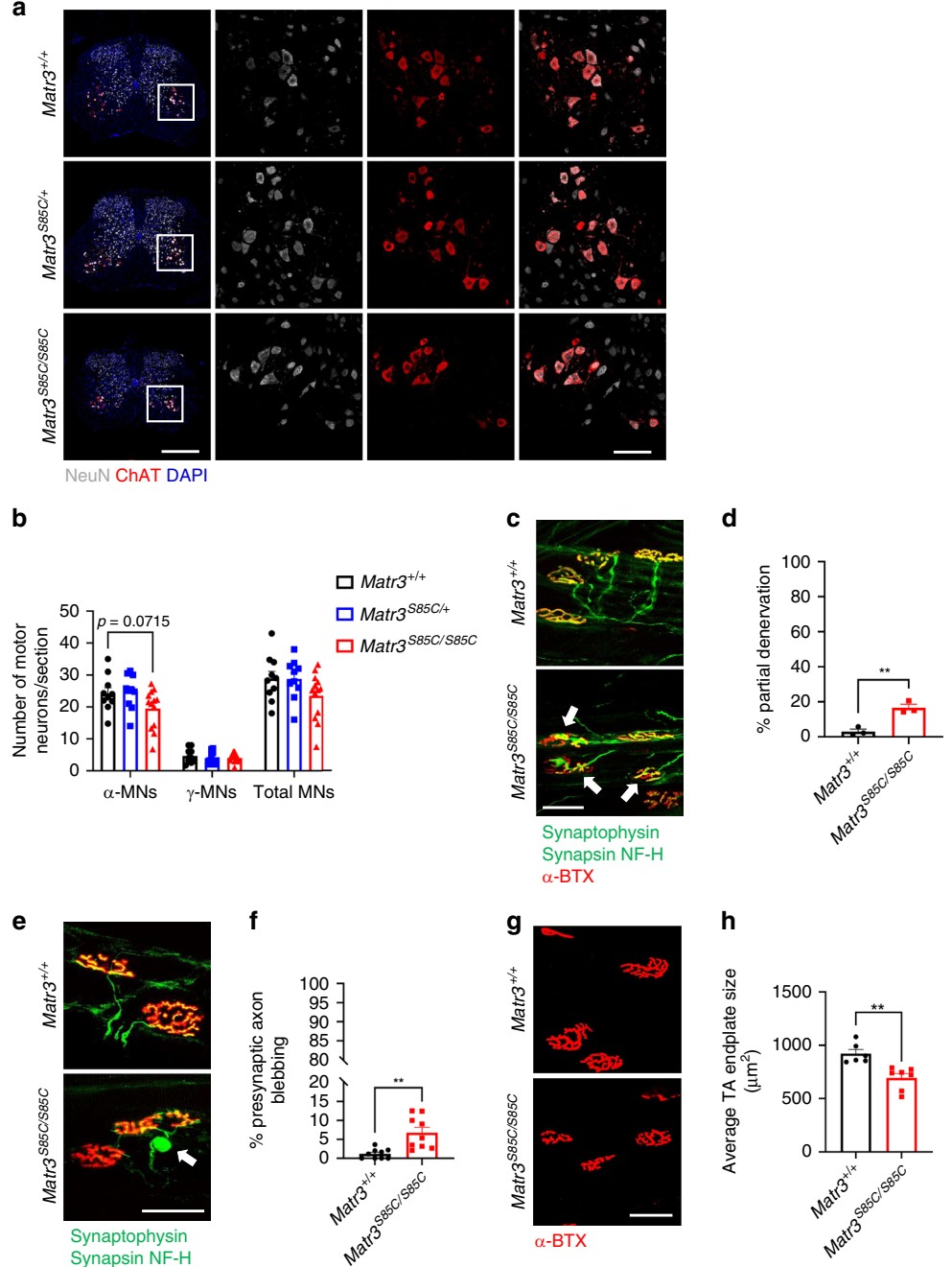

**Fig. 5 Neuromuscular junction defects in homozygous S85C mice at endpoint. a** Representative images of lower lumbar spinal cords (L4–L6) with magnified images showing ChAT-and NeuN-positive α-motor neurons (MNs) and ChAT-positive, NeuN-negative γ-MNs in the ventral horns of the spinal cord. Scale bars indicate 500 μm (left column) and 100 μm (magnified images). **b** Graph shows the number of α- and γ-MNs in the lumbar spinal cord at disease end-stage ($n = 10$ $Matr3^{+/+}$, 10 $Matr3^{S85C/+}$, 13 $Matr3^{S85C/S85C}$). **c** Images showing synaptophysin and synapsin staining of presynaptic terminals, neurofilament H staining of presynaptic axons and α-bungarotoxin (BTX) staining of post-synaptic terminals in the TA muscles at end-stage. White arrows indicate partially denervated NMJs. Scale bar indicates 50 μm. **d** Graph shows the percentage of partially denervated NMJs ($n = 3$ $Matr3^{+/+}$, 3 $Matr3^{S85C/S85C}$, **$p = 0.0053$). **e** Representative images showing synaptophysin and synapsin staining of presynaptic terminals, neurofilament H staining of presynaptic axons and α-bungarotoxin staining of post-synaptic terminals in the TA muscles at end-stage. White arrow indicates axonal bleb proximal to the NMJ. Scale bar indicates 50 μm. **f** Graph shows the percentage of neurofilament-positive axon swelling ($n = 9$ $Matr3^{+/+}$, 9 $Matr3^{S85C/S85C}$, **$p = 0.0017$). **g** Representative images showing α-bungarotoxin staining of endplates in the TA muscles at end-stage. Scale bar indicates 60 μm. **h** Graph shows average TA endplate size ($n = 6$ $Matr3^{+/+}$, 7 $Matr3^{S85C/S85C}$, **$p = 0.0019$). Data shown in **b**, **d**, **f** and **h** are displayed as mean ± SEM and each dot represents a single animal. Significance was determined by unpaired two-tailed $t$-test. Source data are provided as a Source Data file.

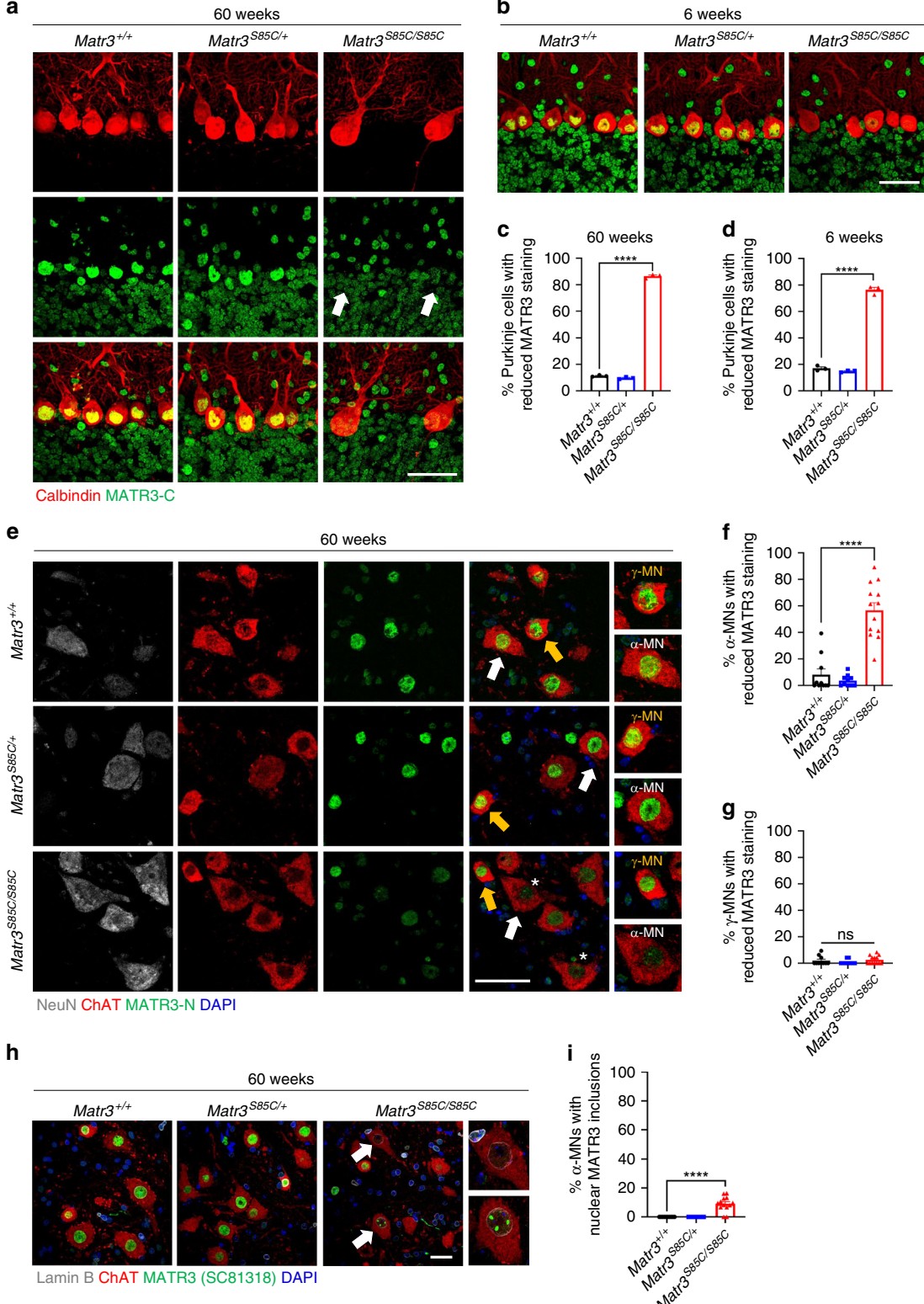

In contrast to significant differential gene expression changes observed in the cerebellum and spinal cord of *Matr3^{S85C/S85C}* mice vs. *Matr3^{+/+}* mice, no significant changes in gene expression were detected in the cortex (Fig. 7c).

Next, we validated whether inflammatory genes identified in RNA-seq data are upregulated in *Matr3^{S85C/S85C}* mice through reverse transcriptase-PCR (RT-PCR). As expected, inflammatory genes including *Clec7a*, *Ccl6*, *Mpeg1* and *Lpl* were significantly

upregulated in the cerebellum of 8–10-week-old *Matr3^{S85C/S85C}* mice compared to wild-type control littermates and this was confirmed in both females and males (Supplementary Fig. 14). Furthermore, we investigated whether *Matr3^{S85C/S85C}* mice show increased activated microglia and astrogliosis in the cerebellum through immunostaining. Indeed, we found significantly increased levels of IBA1-positive microglia and striking astrogliosis (marked by Glial fibrillary acidic protein (GFAP)) in the

**Fig. 6 Reduced MATR3 immunoreactivity in the nucleus of Purkinje cells and α-motor neurons in homozygous S85C knock-in mice. a, b** Representative images showing MATR3 (green) localization and levels (detected with MATR3-C antibody) in the calbindin-positive Purkinje cells (red) and other cells in the cerebellum in mice at (**a**) disease end-stage and at (**b**) 6 weeks. Scale bars indicate 30 μm. White arrows in **a** indicate MATR3 loss in Purkinje cells in homozygous S85C mouse. **c, d** Graphs show the percentage of calbindin-positive Purkinje cells with reduced MATR3 staining in **c** at disease end-stage ($n =$ 3 $Matr3^{+/+}$, 3 $Matr3^{S85C/+}$, 3 $Matr3^{S85C/S85C}$, ****$p < 0.0001$) and in **d** at 6 weeks old ($n = 3$ $Matr3^{+/+}$, 3 $Matr3^{S85C/+}$, 3 $Matr3^{S85C/S85C}$, ****$p < 0.0001$). **e** Representative images showing MATR3 (green) staining (detected with MATR3-N antibody) in ChAT-positive only γ-MNs or ChAT-positive and NeuN-positive α-MNs in the ventral horn of the lumbar spinal cord at disease end-stage. Yellow arrow indicates γ-MN and white arrow indicates α-MN. White asterisks indicate α-MNs with reduced MATR3 staining. Scale bar indicates 50 μm. **f, g** Graphs show the percentage of **f** α-MNs ($n = 10$ $Matr3^{+/+}$, 10 $Matr3^{S85C/+}$, 13 $Matr3^{S85C/S85C}$, ****$p < 0.0001$) and **g** γ-MNs ($n = 10$ $Matr3^{+/+}$, 10 $Matr3^{S85C/+}$, 13 $Matr3^{S85C/S85C}$, ns = not significant) with reduced MATR3 staining. **h** Images showing MATR3 staining (detected by SC-81318 antibody) in ChAT-positive MNs in the ventral horn of the lumbar spinal cord at disease end-stage. White arrows and insets show MNs with reduced MATR3 staining or MATR3-positive nuclear inclusions. Scale bar indicates 30 μm. **i** Graph shows the percentage of α-MNs with nuclear MATR3 inclusions ($n = 10$ $Matr3^{+/+}$, 10 $Matr3^{S85C/+}$, 13 $Matr3^{S85C/S85C}$, ****$p < 0.0001$). Data shown in **c, d, f, g,** and **i** are displayed as mean ± SEM and each dot represents a single animal. Significance was determined by unpaired two-tailed t-test. Source data are provided as a Source Data file.

---

cerebellum of $Matr3^{S85C/S85C}$ mice compared to wild-type mice at 30 weeks of age (Fig. 7f–h). Likewise, significantly increased levels of IBA1-positive microglia were observed in the lower lumbar spinal cord of $Matr3^{S85C/S85C}$ mice (Supplementary Fig. 15). Taken together, these results suggest that neuroinflammation contributes to disease progression in $Matr3^{S85C/S85C}$ mice.

## Discussion

About a dozen missense mutations in *MATR3* have been recently identified in ALS patients[22–28]. Among them, S85C is the most frequently found mutation in *MATR3* and it is associated with earlier disease onset. A few studies have demonstrated the pathogenic potential of the S85C mutation in *MATR3* in cells and animal models[32,34,46,50,51]. However, unraveling the mechanism by which the S85C mutation causes disease has been challenging using overexpression systems and transgenic animal models.

To circumvent the confounding effects of overexpression of exogenous MATR3 and to investigate the impact of an ALS-linked mutation at the physiological level in mice, we introduced a single amino acid change, S85C, in the endogenous *Matr3* allele using the CRISPR/Cas9 system, hereby establishing MATR3 S85C knock-in mice. The S85C mutation does not affect embryonic or early postnatal development. Aging significantly influenced the pathogenic behavior of MATR3 S85C as homozygous S85C mice began to show motor impairments starting at 10 weeks of age and the motor phenotypes progressively worsened over time. These mutant animals show Purkinje cell loss and NMJ defects. Strikingly, MATR3 S85C protein was specifically lost in these neurons. Our neuropathological data suggest that the S85C mutation induces toxicity in at least two neuronal populations, the α-motor neurons and Purkinje cells, and that loss of MATR3 function is involved in disease pathogenesis. In addition, given the presence of MATR3 S85C aggregates in subsets of motor neurons, a toxic gain of MATR3 function could also contribute to neuronal defects. This idea is supported by evidence of an increasing trend in the level of insoluble MATR3 S85C in the cerebellum (Supplementary Fig. 16), which is similar to previous findings[29,32,52]. The timeline of disease progression of homozygous S85C mice is represented in Fig. 8.

Homozygous S85C mice show neuropathological features reminiscent of early-stage ALS including partial denervation associated with axonal defects near the NMJs and smaller end-plate sizes. However, no significant loss of motor neuron cell bodies was found in the lumbar spinal cord. Our results are reminiscent of findings from a very recent study showing NMJ disruption in a *FUS*-ALS mouse model at the early disease stage[8], and supports the "dying-back" hypothesis[4–8]. The NMJs of the motor neurons receive metabolic support and signaling input from the cell bodies. Our observation of loss of mutant MATR3

in the cell bodies of motor neurons suggest that NMJ defects could be due to aberrant changes within the cell body. This model would enable us to explore the cause and mechanisms underlying motor neuron degeneration at the earliest steps in the disease process.

Although motor neurons themselves could contribute to NMJ defects, previous studies have demonstrated that toxicity from skeletal muscle fibers may also contribute to these defects[8,53,54]. Our findings of smaller NMJ endplates, muscle atrophy and reduced muscle strength may suggest muscle contribution to disease pathogenesis. However, our observation of unaltered MATR3 levels in the nuclei of TA muscles suggest that the S85C mutation may not impact muscle tissue. Our finding of loss of MATR3 in the motor neurons favors the hypothesis of cell autonomous contribution of motor neurons to NMJ defects and subsequent muscle pathology and functional impairment. However, this hypothesis should be tested in future studies.

This is the first report to unravel the role of mutant MATR3 in Purkinje cells, as we discovered that Purkinje cells are sensitive to MATR3 S85C toxicity. The cerebellum sizes of the homozygous S85C mice were significantly decreased at disease end-stage, which could be a secondary consequence of Purkinje cell degeneration. Purkinje cells and the associated cerebellar changes may underlie motor coordination defects in homozygous S85C mice. In support of our findings, recent studies from ALS patients and mouse models provide evidence that Purkinje cells and the cerebellum are also targeted[40]. ALS patients with mutations in *ATXN2* or *C9orf72* show a significant loss of Purkinje cells[55,56]. Mouse models of *SOD1*, *C9orf72*, *ATXN2* and *TBK1* exhibit neuropathological changes in Purkinje cells[42,57–60]. In addition, ubiquitinated TDP-43 and p62-positive inclusions were observed in the cerebellum of ALS patients[56,61–67]. Furthermore, cerebellar atrophy was observed in ALS patients[56,68–70]. Our findings, together with recent studies, suggest the involvement of the cerebellum in ALS pathogenesis and calls for clinical studies investigating whether ALS patients diagnosed with MATR3 S85C mutation show signs of cerebellar dysfunction.

Our results suggest that loss of MATR3 function may contribute to neuronal defects. Although further work is required to understand how MATR3 S85C is lost, we propose several hypotheses. One possibility is that the S85C mutation could alter MATR3 protein structure and lead to altered protein properties (e.g., solubility and stability). Although the structure of the RNA recognition domains of MATR3 have been identified, its full structural information and how the S85C mutation might affect MATR3 structure remains to be elucidated. Second, the S85C mutation may lead to aberrant localization of MATR3 and thereby expose MATR3 for protein degradation. The presence of nuclear MATR3 inclusions in the nucleus of some α-MNs

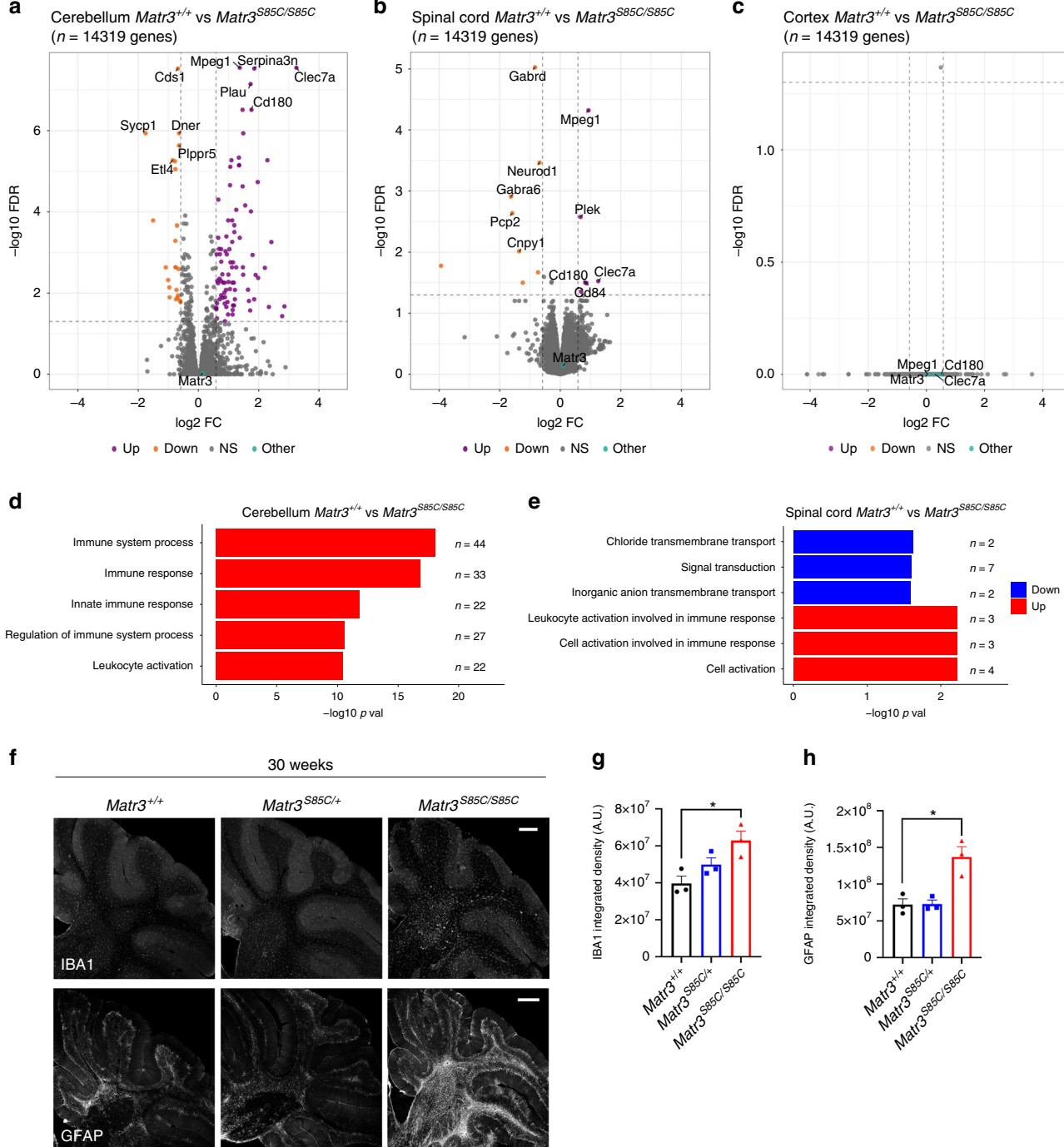

**Fig. 7 Upregulation of immune response genes in homozygous S85C mice at early disease stage. a–c** Volcano plots showing differentially expressed genes in the **a** cerebellum, **b** spinal cord, and **c** cortex of wild-type and homozygous S85C mice at 8–10 weeks old. FDR $p < 0.05$, absolute fold change (abs (FC)) > 1.5. The top five genes with significant expression changes are labeled with gene names. **c** In the cortex, no genes are significantly differentially expressed including immune response genes. **d, e** GO analysis for upregulated (red) or downregulated (blue) genes in the **d** cerebellum and **e** spinal cord. **f** Representative images showing IBA1 and GFAP staining of the cerebellum at 30 weeks of age. Scale bars indicates 300 μm. **g, h** Graphs show the integrated density of **g** IBA1 ($n = 3$ $Matr3^{+/+}$, 3 $Matr3^{S85C/+}$, 3 $Matr3^{S85C/S85C}$, *$p = 0.023$) and **h** GFAP ($n = 3$ $Matr3^{+/+}$, 3 $Matr3^{S85C/+}$, 3 $Matr3^{S85C/S85C}$, *$p = 0.0157$) staining in the cerebellum at 30 weeks old. Data shown in **g** and **h** are displayed as mean ± SEM and each dot represents a single animal. Significance was determined by unpaired two-tailed $t$-test. Source data are provided as a Source Data file.

suggests that MATR3 S85C may be recruited to nuclear bodies, which may be sites of MATR3 protein degradation. Promyelocytic leukemia protein (PML) nuclear bodies in the nuclear matrix are proposed to regulate several cellular processes such as activating and sequestering protein partners as well as degrading unstable proteins[71]. Whether MATR3 S85C localizes to PML

nuclear bodies remains to be tested. Third, increased phosphorylation due to the S85C mutation could contribute to MATR3 degradation. A previous study showed that Protein Kinase A (PKA) phosphorylation of MATR3 following excessive activation of the NMDA receptors in cerebellar neurons resulted in increased MATR3 degradation, which then led to neuron death[72].

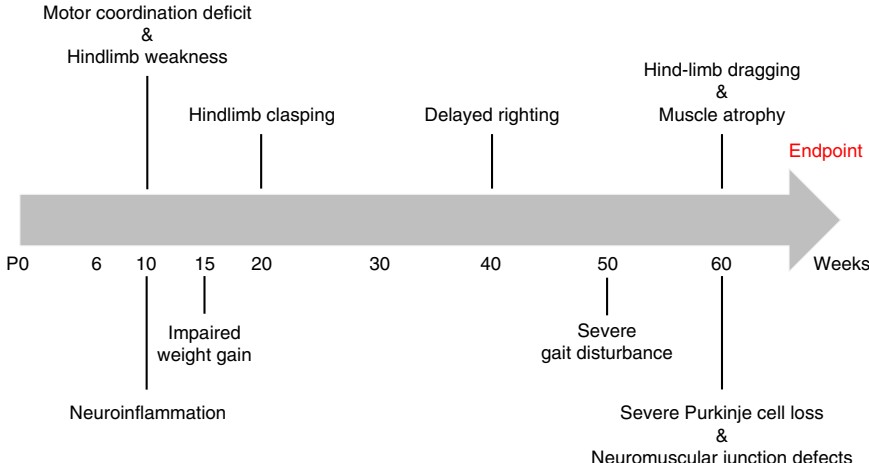

**Fig. 8 Timeline of disease progression of homozygous S85C mice.** Homozygous S85C mice show motor impairments starting at around 10 weeks of age, progressively worsen with age and reach endpoint at around 60 weeks of age. These mutant mice also exhibit neuroinflammation, Purkinje cell loss and NMJ defects.

Therefore, whether the S85C mutation renders MATR3 to be highly phosphorylated by PKA should be tested. Studying the mechanisms by which MATR3 S85C is being lost could help identify potential therapeutic targets for ALS.

*Matr3* knockout mice are perinatal lethal, suggesting that MATR3 plays a critical role during development. Further studies on how *Matr3* knockout mice succumb to death and whether complete absence of MATR3 affects neuronal development would provide insights into ALS pathogenesis. It is important to note that homozygous S85C mice are viable, suggesting that the S85C mutation does not result in a complete loss of MATR3 function during embryonic development. In addition, the S85C mutation does not seem to affect early postnatal development given no obvious difference in weight and behavior compared to the wild-type littermates up to at least 10 weeks of age. As homozygous S85C mice start to show motor phenotypes at 10 weeks of age, aging impacts the pathogenic behavior of MATR3 S85C in Purkinje cells and motor neurons. Previous studies from TDP-43 knock-in mouse models also show age-dependent neurodegeneration. Although one study observed no evidence of motor neuron degeneration in TDP-43 Q331K knock-in mice[73], another study reported evidence of neurodegeneration in TDP-43 M337V or G298S knock-in mice at two and a half years of age[7]. The findings from these TDP-43 studies suggest that aging may trigger the toxic behavior of ALS-linked proteins. Therefore, understanding how aging influences the behavior of MATR3 S85C is key to understanding ALS pathogenesis.

Of note, S85C is an autosomal dominant mutation in humans. Heterozygous S85C mice do not display any behavioral or neuropathological changes seen in the homozygous S85C mice. One explanation is that as the phenotypes are age-dependent, heterozygous S85C knock-in mice may not survive long enough to develop symptoms seen in humans. Another explanation is that humans might be more sensitive to the levels of MATR3 S85C and develop MATR3 S85C-induced neurotoxicity at a lower threshold compared to mice. Although there are mechanistic differences between our MATR3 knock-in mouse model and the human MATR3 S85C disease condition, our mouse model expressing MATR3 S85C under the endogenous mouse regulatory elements serves to model human ALS features which has not been shown before.

Our MATR3 S85C knock-in model will enable us to explore the long-term consequences of an ALS-linked mutation in neurons that are susceptible in ALS. Because aging is a key factor contributing to ALS, investigating genetic or epigenetic changes in aged neurons will provide insights into understanding the role of MATR3 S85C in ALS pathogenesis. In addition, our MATR3 S85C knock-in animals are an excellent model system for exploring the mechanisms underlying selective neuronal vulnerability in ALS. Our findings ask for future research to explore why α-motor neurons and Purkinje cells particularly fail to tolerate MATR3 S85C and lead to neuronal impairment. Performing single-cell RNA sequencing on both affected and non-affected neurons will help us understand this selective vulnerability. Taken together, MATR3 S85C knock-in animals will serve as an important early-stage ALS model for uncovering pathogenic mechanisms, identifying early biomarkers and testing novel therapies.

## Methods

**Generation of *MATR3* knockout and S85C knock-in mice.** All mouse procedures were performed under the approval of the Animal Care Committee (ACC) and housed at The Centre for Phenogenomics (TCP), accredited by the Association for Assessment and Accreditation of Laboratory Animal Care International. Mice were kept on a 12 h light/12 h dark cycle, at 20–22 °C and 43% humidity, with ad libitum access to food and water.

MATR3 S85C knock-in mice were generated by Dr. Lauryl Nutter at TCP using the CRISPR/Cas9 system. To obtain the p.S85C point mutation, the nucleic acid change, c.254 C > G, was introduced into the endogenous *Matr3* mouse locus using guide RNA (gRNA) (5′-TTGCAGTCTATATTTAACAT-3′, Chr18:35572270(+)) and the repair oligonucleotide (5′-AGACAGACCAAAGCT GGCC AAAATATTACTGGCCTGGTCTGTATCTCCACGGTGTTGAGAAG ACAAAGGGAGTGGACCTCTAC TTCCAATaTTgAATATAcACTGCAAATTA TGGGAAGAAGTACTA-3′), which is complementary to the PAM-containing (non-target) DNA strand. To prevent re-cleavage of the repaired allele by Cas9, two silent mutations (c.261T > C/p.F87 and c.264C > T/p.N88) were also introduced. The c.261T > C removes an MseI restriction endonuclease site which was used for genotyping. The CRISPR/Cas9 system and gRNA was introduced into C57BL/6J embryos, which were implanted into pseudo-pregnant females. Germline transmission was verified in a single founder by genotyping and sequencing the F1 pups. Mice were backcrossed to C57BL/6J for four generations to reduce transmission of any off-target mutations, prior to setting up the heterozygous × heterozygous cross to obtain heterozygous, homozygous and wild-type littermates. *Matr3* knockout founders were obtained from non-homologous end-joining using gRNA (5′-TTGCAGTCTATATTTAACAT-3′, Chr18:35572270(+) or 5′-AGTA ATATTTTGGCCAGCTT-3′, Chr18:35572348(+)). Germline transmission was verified by genotyping and sequencing F1 pups, prior to backcrossing to C57BL/6J for four generations and intercrossing.

**Genotyping MATR3 S85C knock-in mice.** Genotyping was performed on tail biopsy DNA from mice using either one or two methods: MseI restriction endonuclease digestion (Fwd: 5′-ACTGCAGCCTTGCTAGTTT-3′; Rev 5′-ATGCGA GGTCTCACCAAAAA-3′) and/or simple allele-discriminating PCR assay

(S85C_Fwd 5′-GCGTTACCATTTTTGAAGCAA-3′; S85C_WT_Rev 5′-CCTC
TACTTCCAATGTTAAATATGG-3′; S85C_PM_Rev 5′-CCTCTACTTCCA
ATATTGAATATGC-3′).

**Genotyping *MATR3* knockout mice**. Genotyping was performed on tail biopsy
DNA from mice using allelic discrimination PCR assay (Fwd: 5′-GGAGCTCAT
AGTGCACTGTCTT-3′; Rev 5′-TTGTCCTCTGGATAGCGACTC-3′).

**Off-target mutation analysis and MATR3 exon sequence verification**. Off-
target analysis was performed on tail biopsy DNA from mice by PCR amplifying
300–600 bp regions surrounding the potential off-target site (Supplementary
Data 1). In addition, the splice sites and exon sequences of mouse *Matr3* were
verified by PCR amplifying 300–600 bp regions (Supplementary Data 2). The PCR
products were cleaned using a commercially available kit (Machery-Nagel) then
submitted for Sanger sequencing at The Centre for Applied Genomics (TCAG) at
the SickKids Research Institute.

**Monitoring of health decline and motor impairment**. Starting at 3 weeks of age,
the body weight of all mice was measured bi-weekly. From 20 weeks of age, animals
were scored weekly for body condition (score 0–14), mobility/activity level (score
0–60), appearance (score 0–25), respiration distress (Y/N) and severe ataxia (>30
min unable to right, Y/N) (Supplementary Data 3).

**Behavioral assays**. Behavioral assays were conducted on different days, so that
only one test was performed per day, allowing at least one overnight rest between
tests. Mice were moved to the behavior testing facility 30 min prior to commencing
the assay to allow the animals to acclimatize to the new environment.

**Rotarod assay**. Rotarod (Ugo Basile, Stoelting) is an automated apparatus that
measures motor coordination, balance and motor learning ability. This assay was
performed similar to a previous study[74]. Mice are placed on an accelerating rotarod
starting at 5 r.p.m. and accelerating to 40 r.p.m. within 300 s. The time (latency) it
takes for the mouse to fall off the rod is measured. The mice were tested four times
a day with a minimal rest period of 15 min between assessments and were tested
for either 2 or 3 consecutive days at 6, 10, 20 and 30 weeks of age.

**Dowel test**. Motor coordination and balance were tested using a 0.9 cm diameter, 1
m-long dowel suspended between two platforms 30 cm above the bench. This assay
was adapted from previous studies[56,75]. The test was conducted with *Matr3*[+/+],
*Matr3*[S85C/+], and *Matr3*[S85C/S85C] littermates at 10, 20 and 30 weeks of age. Mice were
trained to walk across the dowel on the first (training) day. On the second (testing)
day, the mice were placed on the center of the dowel facing one of the platforms and
allowed to move freely. The number of times each mouse crossed the dowel within
two minutes was recorded.

**Open-field assay**. Open-field measures locomotor activity and rearing. The
chamber (43.48 cm × 43.88 cm × 30.28 cm, L × W × H) has three 16-beam IR arrays
on X, Y, and Z axis for both horizontal and vertical movement (Med Associates,
Inc). Mice at 10, 20, and 30 weeks of age were tested. Mice were placed into the
arena near the front wall facing left and allowed to freely explore for 20 min. The
movement was tracked by the number of beams broken and analyzed by the
Activity Monitor (Med Associates, Inc).

**Inverted grid test**. The inverted grid test measures muscle strength. Mice were
placed on a metal grid held approximately 30 cm above the benchtop. After 3 s, the
grid was inverted so that the mice were hanging upsidedown. The time between
grid inversion and when the animal lost its grip was measured and recorded with a
maximum time of 3 min.

**Conditioned fear test**. The conditioned fear test measures aversive memory,
capitalizing on the fact that mice freeze when scared[37,38]. The test was completed
over 2 days in three stages. Training stage: A mouse was placed into a chamber,
after 2 min a sound (85 db) was turned on and after 25 s a electrical shock (0.72
mA) lasting 5 s was administered through a grid. This cycle was repeated once
more, for a total of 5 min. Context stage: The next day the mouse was returned to
the same chamber for 5 min and the length of time the mouse froze was recorded.
Cued stage: At least 2 hours after the context stage, the mouse was placed into a
modified chamber, with different visual and olfactory cues. For 3 min the mouse
was allowed to explore freely (Pre-CS) and then the sound cue (85 db) was played
for 3 min (CS). Freezing time was recorded for each period and compared between
genotypes.

**Delayed righting test**. During weekly scoring, homozygous S85C animals
>20 weeks of age were purposefully placed onto their side in their home cage (with
cage mates present) and timed to determine the length of time required for the

animal to right itself. The time to right for each side was measured and the times
combined determined the delayed righting score.

**Hind-limb dragging**. Three-minute movies taken of freely moving homozygous,
heterozygous S85C, and wild-type animals were analyzed for the presence of hind-
limb dragging. Incidences of hind-limb dragging were counted and compared.

**Footprint test**. The footprint test was used to analyze gait. Fore and hind paws
were painted with non-toxic dyes of different colors (red and blue). The mice were
allowed to freely walk along a paper-covered runway leaving footprints. The hind-
limb stride length (mm) was measured for five consecutive steps, for two replicates
per mouse and three mice per genotype.

**Total protein extraction**. Tissues were homogenized in 10 mL/g ice-cold Urea
buffer (7 M Urea, 2 M Thiourea, 4% CHAPS, 30 mM Tris HCl pH 8.5, PhosSTOP
(Sigma), Protease Inhibitors (Roche)) using a Dounce Tissue Grinder Set (Sigma),
then passed through a 22 G and 26 G needle. The lysate was sonicated at 20 mA for
fifteen 1 s pulses. The samples were mixed with 4× LDS (ThermoFisher Scientific)
and 10× Reducing buffer (ThermoFisher Scientific) prior to western blotting analysis.

**Soluble and insoluble protein extraction**. Soluble and insoluble protein lysates
were obtained following modification of the Malik et al.[32] protocol. Briefly, tissues
were homogenized in 10 mL/g ice-cold RIPA buffer (50 mM Tris HCl pH 7.5,
150 mM NaCl, 1% NP-40, 0.5% Sodium deoxycholate, 0.1% SDS, PhosSTOP
(Sigma), Protease Inhibitors (Roche)) using a Dounce Tissue Grinder Set (Sigma),
then passed through a 22 G and 26 G needle. The lysate was sonicated at 20 mA for
2 min of 5 s pulses. The samples were incubated 30 min on ice prior to ultra-
centrifugation at $100,000 \times g$ for 30 min at 4 °C (rotor: TLA110). The supernatant
(RIPA-soluble fraction) was carefully removed from the pellet and mixed with 4×
LDS (ThermoFisher Scientific) and 10× Reducing buffer (ThermoFisher Scientific),
and boiled for 10 min at 80 °C. Meanwhile, the pellet was washed once with RIPA
buffer prior to resuspending in Urea buffer (7 M Urea, 2 M Thiourea, 4% CHAPS,
30 mM Tris HCL pH 8.5, PhosSTOP (Sigma), Protease Inhibitors (Roche)). The
RIPA-insoluble fraction was then sonicated at 20 mA for fifteen 1 s pulses, then
mixed with 4× LDS (Invitrogen) and 10× Reducing buffer (Invitrogen) prior to
western blotting analysis.

**Western blotting analysis**. Samples were separated on 8% Bis-Acrylamide hand-
cast gels in Tris-Glycine-SDS Running Buffer (25 mM Tris, 192 mM glycine, 0.1%
SDS). The proteins were then transferred onto 0.45 μM Nitrocellulose membrane
(Amersham Protran Premium) in transfer buffer (25 mM Tris, 192 mM Glycine,
10% Methanol), prior to blocking with Licor Odyssey Blocking Buffer (TBS, LiCor)
and probing with primary antibody diluted in blocking buffer. Blots were probed
for MATR3 (Rabbit anti-MATR3-N, HPA036565, 1 : 1000, Sigma; Rabbit anti-
MATR3-C, ab84422, 1 : 1000, Abcam; Mouse anti-MATR3 internal, sc81318, 1 :
1000), and the loading control glyceraldehyde 3-phosphate dehydrogenase
(GAPDH) (mouse anti-GAPDH clone 6C5, CB1001, 1 : 10,000, Calbiochem) prior
to imaging with fluorescent-labeled secondary antibodies (LiCor IRDye 800CW or
IRDye 680RD Goat anti-mouse or Goat anti-rabbit) diluted in TBS using the LiCor
Odyssey Fc imager. Western blottings were quantified using ImageStudio Lite
(v.5.2).

**RNA extraction and RNA-seq analysis**. Tissues (mouse brain cortex divided in
half, cerebellum divided in half, and lumbar spinal cord) were collected from wild-
type, heterozygous, and homozygous littermates at 8–10 weeks of age ($n = 4$ ani-
mals/genotype, only females) and flash frozen in 1 mL of Trizol reagent (Invitro-
gen). The samples were thawed, homogenized, and incubated at room temperature
(RT) for 3 min prior to being mixed with chloroform (Sigma-Aldrich) and cen-
trifuged. The supernatant was moved to a new tube and RNA was precipitated with
isopropanol and the RNA pellet was washed with 75% ethanol. After drying, the
RNA pellet was resuspended in Diethyl pyrocarbonate DEPC-treated water
(Ambion) and warmed for 15 min at 55 °C. Total RNA was stored at −80 °C.

**RNA sequencing**. Before RNA library generation, the RNA was cleaned with
RNeasy MiniElute Cleanup Kit (Qiagen). Total RNA was prepared for mRNA
libraries, which were paired-end sequenced at TCAG (SickKids Research Institute)
on the Illumina NovaSeq S1 flow cell. The average number of reads for each tissue
is as follows: cortex, 25.7 million reads; cerebellum, 76.1 million reads; spinal cord,
77.1 million reads.

**RNA-seq reads processing and quality control**. Read quality was determined
using FastQC (v0.11.7). Adapters were trimmed from reads using trimmomatic
(v0.39) with paired-end default settings, except trimmomatic parameter SLI-
DINGWINDOW:4:15[76]. Quality of trimmed reads was then determined using
FastQC (v0.11.7). Trimmed reads were then mapped to mouse genome (mm10)
using STAR aligner (v2.6.0) with standard ENCODE RNA-seq pipeline options
except "–outFilterMismatchNoverReadLmax 1"[77]. Quality of mapped RNA-seq

reads was performed using QualiMap (v.2.2.1)[78]. GENCODE version M21 was the primary annotation used. Paired-end reads mapping to exonic regions were counted using featureCounts (v1.6.3) with default options, except "-p -B -C -s 2"[79].

**Counts processing and quality control**. R (v3.6.0) was used for downstream analysis of RNA-seq data. Prior to analysis of differential gene expression, read counts were scaled by sample library sizes and read counts per million (CPM) were calculated using edgeR functions cpm() and calcNormFactors()(edgeR v3.26.5)[80,81]. Only genes with CPM ≥ 2 in at least ten samples were retained for downstream analysis. Principal component analysis plots were performed on CPM values using R function prcomp() (ggfortify v0.4.9).

**Differential gene expression analysis**. For detection of DEGs between different mutation genotypes, edgeR (v3.26.5) was used[81]. Quasi-likelihood F-test method was used to test for differential expression (|log2FC| > log2(1.5), FDR-corrected p-value < 0.05) between different genotypes within a given tissue. For every genotype comparison, the comparison formula is (mutant) − (control).

**Functional enrichment analysis**. gProfileR(v0.6.7) was used in R (v3.6.0) to access functional enrichment of DEGs using a list of detected genes (CPM ≥ 2 in ≥10 samples) as statistical background and terms with a minimum set size of 3[82]. Only functional terms passing FDR p-value threshold < 0.05 from GO : BP were considered.

**RNA extraction, cDNA generation, and RT-PCR**. Whole brain or spinal cord from 3-week-old mice was isolated and stored at −80 °C in Trizol reagent. RNA was extracted using the chloroform method detailed above. Two micrograms of total RNA from each sample was used to generate cDNA using random hexamer primers and the M-MLV reverse transcriptase (Invitrogen) according to the manufacturer's directions. PCR was performed using the KOD Hot Start DNA Polymerase (EMD Millipore 710864). For RT-PCR, three animals per genotype were tested with Matr3 primers (Fwd: 5′-CTGGCCCCTTACAAGAGAGA-3′; Rev 5′-CACTGGCTTGCCAAACACTA-3′) and Gapdh primers (Fwd: 5′-ACTCCACT CACGGCAAATTC-3′; Rev 5′-CCTTCCACAATGCCAAAGTT-3′).

For validation of upregulated inflammatory genes, cerebellum from 8- to 10-week-old male or female mice was isolated. RNA was extracted using the chloroform method detailed above. 2 µg of total RNA was used to generate cDNA using 1 : 1 ratio of random hexamer primers and oligo dT primers, and the M-MLV reverse transcriptase (EMD Millipore 710864). PCR was performed using the KOD Hot Start DNA Polymerase (EMD Millipore 710864). The following primers were used for PCR: Clec7a (Fwd: 5′-CTTGCCTTCCTAATTGGATC-3′; Rev 5′-GCATTAATACG GTGAGACGATG-3′), Ccl6 (Fwd: 5′-AAGAAGATCGTCGCTATAACCCT-3′; Rev 5′-GCTTAGGCACCTCTGAACTCTC-3′), Mpeg1 (Fwd: 5′-CAGTCGTC TGGAATGTAAAAAG-3′; Rev 5′-GACTGTGCATTTGTCATAGGG-3′), and Lpl (Fwd: 5′-ATGGATGGACGGTAACGGGAA-3′; Rev 5′-CCCGATACAACCAG TCTACTACA-3′). Expression changes were quantified using ImageJ (v. 1.52p).

**Tissue preparation for immunohistochemical analysis**. Mice were deeply anesthetized with intraperitoneal injection of 100/10 mg ketamine/xylazine per kg body weight. They were transcardially perfused with 1× phosphate-buffered saline (PBS) and then 4% paraformaldehyde (PFA) in PBS solution (PFA/PBS). Following perfusion, the brain, spinal cord column, and TA muscles from both hind limbs were dissected. A sagittal cut was made down the midline of the brain. The brain halves and whole spinal cord column were post-fixed in 4% PFA/PBS at RT for 48 h, then washed with 1× PBS. The spinal cord was removed from the spinal column and the lower lumbar region (L4–L6) was embedded in paraffin. Eight-micrometer transverse serial sections of the spinal cord were cut on a microtome. One brain half was cryoprotected in 30% sucrose before embedding in Optimal Cutting Temperature compound (O.C.T.) and the other was embedded in paraffin. Half brains embedded in O.C.T. were cryosectioned at a thickness of 50 µm and half brains in paraffin were sectioned at 5 µm using a microtome. TA muscles for whole-mount immunofluorescent staining were post-fixed in 4% PFA/PBS for 10 min at RT, then washed with 1× PBS.

**Tissue preparation for H&E-stained muscles**. Mice were anesthetized with iso-flurane gas, then sacrificed by cervical dislocation. The TA and GA muscles from both hind limbs were isolated, then flash frozen in isopentane in liquid nitrogen before being stored at −80 °C until ready to process.

**Immunofluorescence on the brain and spinal cord paraffin sections**. Heat-induced epitope retrieval (HIER) was performed with sodium citrate buffer (pH 6.0) on paraffin spinal cord and brain sections. Sections were incubated with blocking solution (10% normal donkey serum and 0.25% Triton X-100 in 1× PBST) for 1 h at RT. Sections were then incubated with primary antibody diluted in blocking buffer overnight at 4 °C. The following primary antibodies were used: anti-MATR3 antibodies (rabbit polyclonal, ab84422, 1 : 400; rabbit polyclonal, HPA036565, 1 : 200; mouse monoclonal, sc81318, 1 : 100), anti-ChAT antibody (goat polyclonal, ab144P, 1 : 200), anti-lamin B1 antibody (rabbit polyclonal,

Cedarlane 12987-1-AP, 1 : 200), anti-NeuN antibody (mouse monoclonal, mab377, 1 : 200), and anti-calbindin D28k antibody (mouse monoclonal, SWant-300, 1 : 300). Sections were washed with 1× PBST (0.05% Tween-20), then incubated with secondary antibody (488/555/647 Alexa Fluor donkey anti-rabbit/mouse/goat IgG (H + L), 1 : 500 from ThermoFisher Scientific) and DAPI (Millipore D6210, 1 : 1000) for 1 h at RT. Sections were washed with PBST and mounted with Prolong Gold Antifade mountant (ThermoFisher Scientific, P36930). For Purkinje cell axon staining in the cerebellum, HIER was performed with citrate-based antigen unmasking solution (Vector Laboratories, H-3300) on paraffin brain sections, then incubated with blocking solution (5% NDS and 0.3% Triton X-100 in 1× PBS) for 20 min at RT. Sections were then incubated with primary antibodies anti-myelin PLP antibody (rabbit polyclonal, ab28486, 1 : 1000) and anti-neurofilament (mouse monoclonal, BioLegend 837904, 1 : 500) overnight at 4 °C. Sections were washed with PBST, then incubated with secondary antibody (488 Alexa Fluor donkey anti-rabbit and 555 Alexa Fluor donkey anti-mouse, 1 : 1000 from Invitrogen) for 2 h at RT. Sections were washed twice with PBST and once with PBS. Autofluorescence quenching was performed using the TrueVIEW™ Autofluorescence Quenching Kit (Vector Laboratories, SP-8400). Sections were counterstained with DAPI and mounted using VECTASHIELD Vibrance Antifade Mounting Medium. Confocal microscopy images were taken using Zeiss LSM700.

**Immunofluorescence on frozen brain sections**. Fifty-micrometer free-floating brain sections were washed twice with 1× PBS prior to incubating with blocking solution (5% normal donkey serum and 0.3% Triton X-100 in 1× PBS) for 1 h at RT. Sections were then incubated in primary antibody diluted in blocking solution for 48 h at 4 °C. The following primary antibodies were used: anti-IBA1 antibody (rabbit polyclonal, Wako-01919741 1 : 1000) and anti-GFAP antibody (rabbit polyclonal, ab7260 1 : 500). Sections were rinsed with 1× PBS and then incubated with secondary antibody (488 Alexa Fluor donkey anti-rabbit IgG (H + L), 1 : 500 from ThermoFisher Scientific) and DAPI (Millipore D6210, 1 : 1000) for 48 h at 4 °C. Sections were rinsed and then mounted with Prolong Gold Antifade mountant (ThermoFisher Scientific, P36930). All slides were imaged using SP8 Leica confocal microscope (Leica Microsystems, Wetzlar, Germany).

**Chromogenic immunohistochemistry staining**. HIER was performed with sodium citrate buffer (pH 6.0) on paraffin spinal cord and brain sections. Endo-genous peroxidases were blocked by incubating with 30% $H_2O_2$ for 30 min at RT. Sections were incubated with blocking solution (10% normal donkey serum and 0.25% Triton X-100 in 1× PBS) for 1 h at RT, then incubated with primary anti-body diluted in blocking buffer overnight at 4 °C. The following primary antibody was used: anti-MATR3 antibodies (rabbit polyclonal, ab84422, 1 : 400). Sections were washed with 1× PBST (0.05% Tween-20), then incubated with secondary antibody (biotinylated goat anti-rabbit IgG, 1 : 200 from Vector Laboratories) for 1 h at RT. Sections were washed, then incubated with ABC solution (Vectastain ABC HRP Kit, PK-4000) for 1 h at RT. Sections were washed again, then incubated with NovaRED substrate (NovaRED Peroxidase Substrate Kit, Vector Laboratories) for 3 min. Sections were immediately counterstained with hematoxylin and mounted with Permount mounting medium (Fisher Scientific, SP15-100).

**Whole-mount immunostaining of TA muscles**. TA muscles were teased into thin pieces before being permeabilized with 2% PBST for 30 min, then blocked with blocking solution (4% bovine serum albumin and 1% Triton X-100 in 1× PBS) for 30 min at RT. The tissues were then incubated in primary antibody diluted in blocking solution overnight at 4 °C. The following primary antibodies were used: anti-synaptophysin antibody (rabbit polyclonal, Synaptic Systems 101 002, 1 : 200), anti-synapsin I antibody (rabbit polyclonal, ab64581, 1 : 200), and anti-neurofilament H antibody (chicken polyclonal, ab4680, 1 : 200). After washing, tissues were incubated with secondary antibody (488 Alexa Fluor donkey anti-rabbit or 488 Alexa Fluor goat anti-chicken, 1 : 250 from ThermoFisher Scientific), α-bungarotoxin (Alexa Fluor 555 conjugate, Invitrogen B35451, 1 : 500), and DAPI (Millipore D6210, 1 : 1000) for 1 h at RT. Tissues were then washed and mounted with Prolong Gold Antifade mountant (ThermoFisher Scientific, P36930). Slides were imaged using SP8 Leica confocal microscope (Leica Microsystems, Wetzlar, Germany).

**H&E staining of TA and GA muscles**. Eight-micrometer frozen muscle sections were stained with Harris' hematoxylin for 8 min (Millipore, HHS32) and excess hematoxylin was removed with 1% acid alcohol (ten dips). The sections were rinsed with tap water and dipped in 95% ethanol before counterstained with Eosin Y/Phloxine B (ten dips). Excess Eosin-Phloxine staining was removed with two changes of 95% ethanol dehydrated with 100% ethanol. Sections were then cleared with two changes of xylene and mounted with Permount mounting medium (Fisher Scientific, SP15-100).

**Motor neuron quantification**. Four 8 µm-thick serial sections from L4–L6 of the lumbar spinal cord were sectioned and stained. A confocal z-series was imaged for each section at ×40 magnification, 0.75 zoom with a step size of 0.5 µm. Motor neurons in the ventral horn that were ChAT-positive and NeuN-positive (α-motor neurons) or ChAT-positive and NeuN-negative (γ-motor neurons) were manually

counted. The percentage of ChAT-positive and NeuN-positive or ChAT-positive and NeuN-negative ventral horn motor neurons with reduced MATR3 staining (reduced compared to staining in surrounding cells) or nuclear MATR3 inclusions were also quantified manually. Four spinal cord sections from 10 $Matr3^{+/+}$, 10 $Matr3^{S85C/+}$, and 13 $Matr3^{S85C/S85C}$ mice were used for quantification.

**Cerebellum size and Purkinje cell quantification.** Four 5 μm-thick serial sections of the brain were sectioned and stained. A confocal z-series of the cerebellum was imaged for each section at ×5 magnification, 0.75 zoom with a step size of 0.5 μm. The whole cerebellum was outlined and the area of the cerebellum was measured using ImageJ. The number of calbindin-positive Purkinje cells in the whole cerebellum was manually counted using ImageJ. The percentage of Purkinje cells with reduced MATR3 staining (compared to staining in surrounding cells, i.e., cells in the molecular and granular layer) was also manually quantified. Three animals per genotype were counted.

**Quantification of IBA1 and GFAP staining.** Four 50 μm-thick serial sections of the brain and four 8 μm-thick serial sections of the lower lumbar spinal cord were sectioned and stained. A confocal z-series of the cerebellum was imaged for each section at ×5 magnification, 0.75 zoom with a step size of 5 μm. The integrated densities of IBA1 and GFAP staining were quantified after converting the images to grayscale and subtracting background using default settings (50 pixels) in ImageJ. Four sections from three mice per genotype were quantified.

**NMJ quantification.** A confocal z-series of randomly selected regions in the whole TA muscle from both legs were imaged at ×40, 0.75 zoom with a step size of 3 μm to capture at least 100 NMJs per animal. Over 50 NMJs per animal were visibly connected to presynaptic axons and, out of these, the number of NMJs with presynaptic axon swelling was manually counted. Nine $Matr3^{+/+}$ and nine $Matr3^{S85C/S85C}$ mice were used for quantification. Endplates marked by 555 Alexa Fluor-conjugated α-bungarotoxin were outlined and their areas were quantified using ImageJ. Over 100 NMJs per animal was used to quantify endplate sizes. Six $Matr3^{+/+}$ and seven $Matr3^{S85C/S85C}$ mice were used for quantification. Out of the NMJs that were visibly connected to presynaptic axons, endplates showing <80% and >1% overlap with presynaptic terminal staining were classified as partially denervated. Three $Matr3^{+/+}$ and three $Matr3^{S85C/S85C}$ mice were used for quantification.

**Muscle fiber size quantification.** H&E-stained TA and GA muscle sections were imaged using a Leica DM2000 microscope at ×20 magnification. For each animal, muscle fibers were outlined and their areas quantified using ImageJ. Four TA muscle sections from eight $Matr3^{+/+}$, seven $Matr3^{S85C/+}$, and seven $Matr3^{S85C/S85C}$ mice, as well as four GA muscle sections from four $Matr3^{+/+}$, five $Matr3^{S85C/+}$, and five $Matr3^{S85C/S85C}$ mice were used for quantification.

**Statistical analysis and reproducibility.** Statistical significance was determined using unpaired, two-tailed Student's $t$-tests or two-way analysis of variance with Dunnett's correction for multiple comparisons, as indicated. All statistical tests were performed using GraphPad Prism 7.00 software package and data are displayed as mean ± SEM, with each datapoint representing an animal: *$p < 0.05$, **$p < 0.01$, ***$p < 0.001$, ****$p < 0.0001$, ns = not significant. Unless noted otherwise, at least three biological replicates per genotype were tested for RT-PCR, western blotting, and immunohistochemistry results.

**Reporting summary.** Further information on research design is available in the Nature Research Reporting Summary linked to this article.

## Data availability

Authors can confirm that all relevant data are included in the paper and its Supplementary Information. Source data are provided with this paper. Gene expression data (accession code: E-MTAB-8838) has been deposited in the ArrayExpress database[47]. The raw immunofluorescence and immunohistochemistry images generated and/or analyzed during the current study are available in the figshare repository https://doi.org/10.6084/m9.figshare.12824312[83]. All other relevant data are available from the authors upon request. Source data are provided with this paper.

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

## Acknowledgements

We thank the members of the Park laboratory for insightful discussions on this project. We acknowledge Drs. Freda Miller and Johanna Rommens for helpful comments on the manuscript. J.P. is grateful to Dr. Huda Zoghbi for her generosity for giving freedom and financial support to initiate the MATR3 project while in the Zoghbi laboratory, and Dr. Jiyoen Kim in the Zoghbi lab for insightful discussions and invaluable help with experimental methodologies. We thank Hee Jin (Hayley) Shin for help with off-target sequencing analysis. We thank Terence Gall Duncan from Dr. Christopher Pearson lab for help, and Dr. Zhenya Ivakine and Dr. Dwi Kemaladewi from Dr. Ronald Cohn lab for experimental methodologies. We also thank the advice and support of Dr. Monica Justice and Dr. Lauryl Nutter for the generation of MATR3 S85C knock-in mice. We appreciate the support from TCP for mouse care and maintenance, mutant generation, histology, and behavioral phenotyping, from TCAG for RNA sequencing and from SickKids imaging facility for the use of microscopes. This study was supported by ALS Canada-Brain Canada career transition award, ALS Canada Project grant, The Scottish Rite Charitable Foundation of Canada, Canada Research Chairs program, and SickKids start-up funds to J.P. Q.T. receives support from the Natural Sciences and Engineering

Research Council of Canada (NSERC) and the Canada Research Chairs program. M.D.W. is supported by the Canada Research Chairs Program and an Early Researcher Award from the Ontario Ministry of Research and Innovation. C.C. was supported in part by NSERC grant RGPIN-2019-07014 to M.D.W. C.S.K. and C.C. were supported by SickKids Restracomp Scholarship. M.Z. is an awardee of Ontario Graduate Scholarship. M.K. is an awardee of Canada Graduate Scholarship.

## Author contributions

C.S.K., R.v.B., J.R.K., X.X.L.C., J.L., W.I.C., M.Z., C.A., K.M., Q.T., and J.P. designed and performed the experiments and curated the data. C.S.K., R.v.B., J.R.K., X.X.L.C., M.K., and J.P. analyzed and interpreted the results. C.C. performed the bioinformatic analysis and M.D.W. supervised and interpreted the analysis. C.S.K., R.v.B., J.R.K., X.X.L.C., and J.P. wrote the manuscript. All authors edited, read, and approved the manuscript. J.P. supervised the project.

## Competing interests

The authors declare no competing interests.
