## [Peer Review File · Nature Communications]

Reviewers' comments:

Reviewer #1 (Remarks to the Author):

The authors describe the phenotypic and neuropathological features of a MATR3 S85C knock out mouse. Strengths of this manuscript include the novelty of the mouse model, the use of CRISPR to generate an endogenously expressed mutant, the depth of the phenotyping and neuropathological analysis performed, and the clear presentation style of the manuscript. The involvement of the cerebellum in these animals is notable. There are no significant weaknesses.

Additional comments and suggestions

- Aside from the muscle spindles, was the muscle evaluated in these animals? It would be good to know if there was evidence of myopathy in addition to motor neuron degeneration.
- The authors suggest that their data support the hypothesis that the motor neuron degeneration occurs because of issues at the neuromuscular synapse. However, this reviewer wonders if their data support the position that the motor neurodegeneration originates in the cell bodies of the neurons. The neuromuscular junction of the motor neuron is a metabolically active, relying heavily on the cell body to support it. It is possible that, as the cell body runs into problems, it is the most distal parts that are the first to feel the effects. Under this paradigm, the first visible sign of the issues would be in the distal neuromuscular junction giving rise to an apparent dying back observation. However, this is an effect, not a cause. Supporting this notion is the observation of changes in expression of MATR3 in the cell bodies of the presymptomatic animals, and the fact that Purkinje cells (that lack muscle contact) are also degenerating. This issue can be dealt with efficiently by the authors in the Discussion and does not require changes to the Results or Methods section.
- Are there plans to make the animals be made publicly available through JAX? They would be an excellent resource.

Reviewer #2 (Remarks to the Author):

MATRIN3 mutations are causative of amyotrophic lateral sclerosis (ALS) together with distal myopathy. The manuscript by Kao, Van Bruggen, Kim, Chen et al provide novel insights into the role of Matrin 3 in neurodegeneration by generating and characterising the first Knock In (KI) mouse model carrying a pathogenic Matr3 mutation.

The characterisation of a new mouse model is a challenging endeavour and in general, the work presented here is of good quality and might be suitable for publication if several issues could be addressed. New disease models are as good as the new findings they allow us to discover. In this case, perhaps the major finding is the identification of the specific degeneration of Purkinje cells in the cerebellum of Matr3 mutant mice. However, the mice do not develop motor neuron degeneration, which is the defining feature of ALS.

There are a number of major issues that will need to be addressed:

1. Throughout the manuscript, and even in the title, the authors suggest that the new mouse strain model ALS. However, the defining feature of ALS is the degeneration of motor neurons, and there is no motor neuron degeneration reported, only a non-significant reduction of alpha motor neurons. The authors report Matrin3 pathology in alpha but not gamma motor neurons together with a reduction in end plate size from the TA muscle; however, it is not clear how these features could impact the motor phenotype. Overall, with the data presented, the motor abnormalities developed by homozygous mutant mice are likely to be chiefly explained by the strong Purkinje cell degeneration. Thus, although the model will be clearly useful to study the role of Matrin 3 in the early stages of the disease, particularly in the cerebellum, with the data presented it is not clear if the motor neurons are involved at all in the development of the motor abnormalities. Thus,

as the mice do not develop motor neuron degeneration, they do not model ALS and efforts should be made throughout the manuscript to acknowledge this major issue.

2. There is particular skeletal muscle involvement in patients carrying MATR3 mutations causing a distal myopathy. However, there is no data presented for the possible muscle involvement in the development of the motor dysfunction in the mutant mice. To address this major issue, extensive muscle pathology will need to be presented at different ages from different muscles, as it is likely that distal muscle pathology might be a contributor towards the motor dysfunction. This should include a more profound characterisation of the neuromuscular junctions, particularly to assess if there is a degree of denervation. Although it may not be possible due to time constraints, it would have been useful to assess muscle strength throughout the progression of the disease in the mutant mice.

3. Cerebellar pathology is present in some ALS cases, prominently C9orf72 carriers, but it is not thought to be a defining feature of the disease. This finding suggests that, at least in the mouse, Matrin 3 is particularly toxic to Purkinje cells. However, what is not clear from the work presented is if this could also be the case in patients carrying MATR3 mutations. I anticipate that analysing the cerebellum of MATR3 mutant carriers might be challenging due to the rarity of MATR3 mutations, but the authors should at least try to validate this key finding in human material.

There are also a number of minor issues that should be addressed, including:

4. Mutations in MATR3 are thought to affect the stability and solubility of the protein. The authors present data western blot data only for young animals but should also assess at least protein levels in aged individuals, particularly as they report Matrin3 pathology.

5. Patients carrying MATR3 mutations also develop dementia. It would have been useful to include some cognitive phenotyping to start to address this issue.

Reviewer #3 (Remarks to the Author):

The manuscript from Kao et al. describes efforts to generate and characterize a novel MATR3 S85C knock-in mouse model for ALS. Prior MATR3 mouse models have used overexpression of a human gene, whereas this knock-in model preserves endogenous expression levels throughout the lifetime of the mouse. The authors detect weight loss, reduced survival, motor defects, motor neuron pathology and Purkinje cell loss in homozygous MATR3 S85C knock-in mice. This study further supports a direct role for this MATR3 mutation as a causative agent in human disease. The supplemental data nicely augments the data presented in the manuscript. While this represents a novel mouse model that recapitulates some features of ALS, there are a number of issues and questions regarding the current manuscript.

1) A crucial issue not explored by the authors is the presence or absence of any muscle pathology in the S85C knock-in mice. The MATR3 S85C mutation has been detected in both ALS and distal myopathy patients. The authors detect neuromuscular junction alterations in the S85C knock-in mice but never examine signs of myopathy over time in the mice.

2) The predominant cell type impacted by the C85C knock-in are Purkinje cells of the cerebellum. Cerebellum pathology has not previously been observed in patients that harbor the MATR3 S85C mutation. The lack of significant loss of spinal cord motor neurons in homozygous mice even at end stage and no evidence of upper motor neuron loss indicates that it does not mimic the neurodegenerative pattern of the human disease. The loss of MATR3 immunoreactivity in motor neurons and Purkinje cells is quite interesting, though the functional consequence of this is unknown. Matrin 3 functions in mRNA splicing and nuclear export, so it is possible that gene

expression changes noted in Figure 7 may be partially due to loss of MATR3 in specific neuronal cell types and/or increased inflammation observed at Day 30.

3) It is unclear why the authors present data on inflammatory markers only at 30 weeks of age, which does not correlate to time points used in other studies. A more in-depth examination of the timing for inflammation in this mouse model is warranted.

4) As noted by the authors, the human MATR3 S85C disease exhibits a dominant genetic effect whereas the heterozygous knock-in mice exhibit no phenotype. This also suggests significant mechanistic differences between the knock-in rodent model and the human disease.

5) The authors often interpret their immunostaining data to indicate a loss or reduction in MATR3 expression (e.g., text describing Figure 6), but the lack of staining may not indicate a loss of MATR3 gene expression but potentially a change in protein turnover or loss of the epitope recognized by the antibody. Additional studies are needed to explore the potential change in MATR3 gene expression in this mouse model. In addition, a number of MATR3 antibodies are used throughout the study (sometimes different antibodies from figure to figure) and it's unclear if similar results are obtained by each of the antibodies.

Reviewer #1

We thank the reviewer for his/her appreciation of this study and strong support for the publication of this manuscript.

Q1) Aside from the muscle spindles, was the muscle evaluated in these animals? It would good to know if there was evidence of myopathy in addition to motor neuron degeneration.

A1) As suggested, we assessed the skeletal muscles in *Matr3*^{S85C/S85C} mice for evidence of myopathy. We found that *Matr3*^{S85C/S85C} mice show visibly reduced size of their hind-limbs (**Supplementary Fig. 12a**). Muscle histology demonstrated that *Matr3*^{S85C/S85C} mice exhibit a significant decrease in muscle fiber sizes of their tibialis anterior (TA) and gastrocnemius (GA) muscles compared to wildtype control mice (**Supplementary Fig. 12b, c, e**). However, we did not find any characteristic myopathic changes such as internalized nuclei, rimmed vacuoles and fibrosis in the muscle tissue of our mice (**Supplementary Fig. 12b, d, f**). These results demonstrate that *Matr3*^{S85C/S85C} mice show muscle atrophy without prominent myopathic features. In addition, we found that MATR3 S85C is predominantly localized in the nuclei of TA muscles in *Matr3*^{S85C/S85C} mice and its staining intensity was not altered compared to wildtype muscles (**Supplementary Fig. 12g**). This result is in contrast to striking MATR3 pathology observed in α -motor neurons (**Fig. 6e-i**). Although muscle atrophy in *Matr3*^{S85C/S85C} mice is reminiscent of mild cases of MATR3 S85C-linked myopathy (Mensch *et al. Experimental Neurology* 2018), lack of prominent myopathic features and unaltered staining levels of MATR3 S85C suggest that muscle atrophy may be secondary to other defects, such as motor neuron defects.

Q2) The authors suggest that their data support the hypothesis that the motor neuron degeneration occurs because of issues at the neuromuscular synapse. However, this reviewer wonders if their data support the position that the motor neurodegeneration originates in the cell bodies of the neurons. The neuromuscular junction of the motor neuron is a metabolically active, relying heavily on the cell body to support it. It is possible that, as the cell body runs into problems, it is the most distal parts that are the first to feel the effects. Under this paradigm, the first visible sign of the issues would be in the distal neuromuscular junction giving rise to an apparent dying back observation. However, this is an effect, not a cause. Supporting this notion is the observation of changes in expression of MATR3 in the cell bodies of the presymptomatic animals, and the fact that Purkinje cells (that lack muscle contact) are also degenerating. This issue can be dealt with efficiently by the authors in the Discussion and does not require changes to the Results or Methods section.

A2) We completely agree with the reviewer that the neuromuscular junction defects could be originating from the cell body dysfunction as our finding shows loss of mutant MATR3 in the cell bodies of motor neurons. We have revised the Discussion section accordingly.

Q3) Are there plans to make the animals be made publicly available through JAX? They would be an excellent resource.

A3) Yes, our mice will be available to the research community. We will deposit the MATR3 S85C

knock-in mice in JAX following publication.

Reviewer #2

We thank the reviewer for critical review and helpful comments and suggestions. We hope the following answers satisfy all of the reviewer's concerns.

Major comments

Q1) Throughout the manuscript, and even in the title, the authors suggest that the new mouse strain model ALS. However, the defining feature of ALS is the degeneration of motor neurons, and there is no motor neuron degeneration reported, only a non-significant reduction of alpha motor neurons. The authors report *Matrin3* pathology in alpha but not gamma motor neurons together with a reduction in end plate size from the TA muscle; however, it is not clear how these features could impact the motor phenotype. Overall, with the data presented, the motor abnormalities developed by homozygous mutant mice are likely to be chiefly explained by the strong Purkinje cell degeneration. Thus, although the model will be clearly useful to study the role of *Matrin 3* in the early stages of the disease, particularly in the cerebellum, with the data presented it is not clear if the motor neurons are involved at all in the development of the motor abnormalities. Thus, as the mice do not develop motor neuron degeneration, they do not model ALS and efforts should be made throughout the manuscript to acknowledge this major issue.

A1) We thank the reviewer for raising a concern of *Matr3^{S85C/S85C}* mouse model's clinical relevance to ALS. As the reviewer suggested in the following comment Q2, we have assessed the innervation status of the motor neurons in the tibialis anterior muscles of *Matr3^{S85C/S85C}* mice. Our results reveal that *Matr3^{S85C/S85C}* mice show a significant percentage of motor neurons that are partially denervated in the muscles compared to wildtype control littermates (**Fig. 5c, d & Supplementary Fig. 9**). This additional evidence of motor neuron denervation provides support that *Matr3^{S85C/S85C}* mouse model is a clinically relevant model for ALS.

Collectively, our results clearly show that *Matr3^{S85C/S85C}* mice exhibit neuromuscular junction defects including partially denervated motor neurons along with axonal swelling and decreased endplate size (**Figure 5**). This is reminiscent of findings from a very recent study showing neuromuscular junction disruption in *FUS*-ALS mouse model at early disease stage (Picchiarelli *et al. Nat Neuroscience* 2019). We acknowledge that these motor neuron phenotypes are early pathogenic events in ALS, so we have revised the title and all sections in the manuscript accordingly.

Q2) There is particular skeletal muscle involvement in patients carrying *MATR3* mutations causing a distal myopathy. However, there is no data presented for the possible muscle involvement in the development of the motor dysfunction in the mutant mice. To address this major issue, extensive muscle pathology will need to be presented at different ages from different muscles, as it is likely that distal muscle pathology might be a contributor towards the motor dysfunction. This should include a more profound characterisation of the neuromuscular junctions, particularly to assess if there is a degree of denervation. Although it may not be possible

due to time constraints, it would have been useful to assess muscle strength throughout the progression of the disease in the mutant mice.

A2) We thank the reviewer for raising this point regarding the relevance of MATR3 S85C knock-in mice to ALS or myopathy. As suggested, we have conducted additional experiments, and our findings strongly suggest that MATR3 S85C knock-in mice is a clinically relevant model for ALS.

First, we examined the integrity of neuromuscular junctions (NMJs) in the tibialis anterior (TA) muscles in *Matr3*^{S85C/S85C} mice at end disease stage. We found a significant increase in the percentage of partially denervated NMJs in *Matr3*^{S85C/S85C} mice compared to wildtype littermates (**Fig. 5c, d & Supplementary Fig. 9**). Our newly added results reveal a significant neuromuscular junction defect in *Matr3*^{S85C/S85C} mice.

Second, we assessed the skeletal muscles in *Matr3*^{S85C/S85C} mice to identify evidence of muscle pathology. We found that *Matr3*^{S85C/S85C} mice show visibly reduced hind-limb size at end disease stage (**Supplementary Fig. 12a**). Muscle histology demonstrated that *Matr3*^{S85C/S85C} mice exhibit a significant decrease in muscle fiber size compared to wildtype control mice (**Supplementary Fig. 12b, c, e**). However, we did not find any prominent myopathic changes such as internalized nuclei, rimmed vacuoles and fibrosis in the muscle tissue of our mice (**Supplementary Fig. 12b, d, f**). These results demonstrate that *Matr3*^{S85C/S85C} mice show muscle atrophy without prominent myopathic features. In addition, we found that MATR3 S85C is predominantly localized in the nuclei of TA muscles in *Matr3*^{S85C/S85C} mice and its staining intensity was not altered compared to wildtype muscles (**Supplementary Fig. 12g**). This result is in stark contrast to striking MATR3 pathology observed in α -motor neurons (**Fig. 6e-i**). Although muscle atrophy in *Matr3*^{S85C/S85C} mice is reminiscent of mild cases of MATR3 S85C-linked myopathy (Mensch *et al. Experimental Neurology* 2018), lack of prominent myopathic features and unaltered staining levels of MATR3 S85C suggest that muscle atrophy may be secondary to other defects, possibly motor neuron defects.

In addition, we have conducted an inverted grip assay to assess muscle strength on *Matr3*^{S85C/S85C} animals. We found that *Matr3*^{S85C/S85C} mice exhibited a significantly reduced muscle strength compared to wildtype littermates at 30 and 55 weeks of age (**Fig. 3d, e**).

Collectively, our additional evidence of motor neuron denervation, muscle atrophy as well as reduced muscle strength support that *Matr3*^{S85C/S85C} mouse model is a clinically relevant model for ALS.

Q3) Cerebellar pathology is present in some ALS cases, prominently C9Orf72 carriers, but it is not thought to be a defining feature of the disease. This finding suggests that, at least in the mouse, Matrin 3 is particularly toxic to Purkinje cells. However, what is not clear from the work presented is if this could also be the case in patients carrying MATR3 mutations. I anticipate that analysing the cerebellum of MATR3 mutant carriers might be challenging due to the rarity of MATR3 mutations, but the authors should at least try to validate this key finding in human material.

A3) We agree with the reviewer that validating cerebellar pathology in human patients would be ideal. Unfortunately, due to the rarity of ALS patients with the MATR3 S85C mutation, we have been unable to obtain patient tissue to validate our findings.

Minor comments

Q4) Mutations in MATR3 are thought to affect the stability and solubility of the protein. The authors present data western blot data only for young animals but should also assess at least protein levels in aged individuals, particularly as they report Matrin3 pathology.

A4) As the reviewer commented, the S85C mutation has been previously shown to reduce MATR3 solubility (Malik *et al. eLife* 2018). As suggested, we collected bulk cerebellar tissue from 60-week-old animals and the protein lysate was then divided into RIPA-soluble and RIPA-insoluble fractions. Interestingly, western blots revealed an increasing trend in MATR3 levels in the RIPA-insoluble fraction in *Matr3^{S85C/+}* and *Matr3^{S85C/S85C}* cerebellum compared to *Matr3^{+/+}* cerebellum (**Supplementary Fig. 16**). This result supports the previous finding that the S85C mutation decreases the protein solubility of MATR3 (Malik *et al. eLife* 2018).

However, the protein levels we observe through western blot analysis do not coincide with the results we observe through immunostaining where MATR3 levels are markedly decreased in Purkinje cells in *Matr3^{S85C/S85C}* cerebellum (**Fig. 6a, c**). We think that this discrepancy is due to testing with bulk cerebellar tissue where the majority of cells are cerebellar granule cells in which MATR3 is not lost.

Q5) Patients carrying MATR3 mutations also develop dementia. It would have been useful to include some cognitive phenotyping to start to address this issue.

A5) A previous study revealed that ALS patients carrying the F115C mutation in *MATR3* exhibited dementia in addition to ALS, while there have been no reported incidences of dementia in patients with the S85C mutation (Johnson *et al. Nature Neuroscience* 2014). However, given that ALS patients often develop dementia, we conducted a memory test on our mouse cohort of *Matr3^{+/+}*, *Matr3^{S85C/+}* and *Matr3^{S85C/S85C}* mice at 40 weeks of age. The contextual and cued fear conditioning test is a classical paradigm that assesses associative fear learning and memory in mice. Interestingly, we found that male *Matr3^{S85C/S85C}* animals showed a significantly reduced time in freezing during both the contextual and cued tests compared to wildtype mice, demonstrating memory deficits in *Matr3^{S85C/S85C}* male mice (**Supplementary Fig. 7g**). However, female *Matr3^{S85C/S85C}* animals exhibited no differences in both contextual and cued tests. This result is unexpected as the S85C mutation has not been associated with male-specific dementia. Further studies are warranted to validate and examine memory deficits in these mice. However, this is beyond the scope of this study.

Reviewer #3

We thank the reviewer for critical review and constructive comments and suggestions. We hope the following answers satisfy all the reviewer's concerns.

Q1) A crucial issue not explored by the authors is the presence or absence of any muscle pathology in the S85C knock-in mice. The MATR3 S85C mutation has been detected in both ALS and distal myopathy patients. The authors detect neuromuscular junction alterations in the S85C knock-in mice but never examine signs of myopathy over time in the mice.

A1) As suggested, we assessed the skeletal muscles in *Matr3*^{S85C/S85C} mice for evidence of myopathy. We found that *Matr3*^{S85C/S85C} mice show visibly reduced size of their hind-limbs (**Supplementary Fig. 12a**). Muscle histology demonstrated that *Matr3*^{S85C/S85C} mice exhibit a significant decrease in muscle fiber sizes of their tibialis anterior (TA) and gastrocnemius (GA) muscles compared to wildtype control mice (**Supplementary Fig. 12b, c, e**). However, we did not find any characteristic myopathic changes such as internalized nuclei, rimmed vacuoles and fibrosis in the muscle tissue of our mice (**Supplementary Fig. 12b, d, f**). These results demonstrate that *Matr3*^{S85C/S85C} mice show muscle atrophy without prominent myopathic features. In addition, we found that MATR3 S85C is predominantly localized in the nuclei of TA muscles in *Matr3*^{S85C/S85C} mice and its staining intensity was not altered compared to wildtype muscles (**Supplementary Fig. 12g**). This result is in contrast to striking MATR3 pathology observed in α -motor neurons (**Fig. 6e-i**). Although muscle atrophy in *Matr3*^{S85C/S85C} mice is reminiscent of mild cases of MATR3 S85C-linked myopathy (Mensch *et al. Experimental Neurology* 2018), lack of prominent myopathic features and unaltered staining levels of MATR3 S85C suggest that muscle atrophy may be secondary to other defects, possibly motor neuron defects.

Q2) The predominant cell type impacted by the S85C knock-in are Purkinje cells of the cerebellum. Cerebellum pathology has not previously been observed in patients that harbor the MATR3 S85C mutation. The lack of significant loss of spinal cord motor neurons in homozygous mice even at end stage and no evidence of upper motor neuron loss indicates that it does not mimic the neurodegenerative pattern of the human disease. The loss of MATR3 immunoreactivity in motor neurons and Purkinje cells is quite interesting, though the functional consequence of this is unknown. Matrin 3 functions in mRNA splicing and nuclear export, so it is possible that gene expression changes noted in Figure 7 may be partially due to loss of MATR3 in specific neuronal cell types and/or increased inflammation observed at Day 30.

A2) As we understand the reviewer's concern about the lack of significant motor neuron loss in *Matr3*^{S85C/S85C} mice, we have evaluated the neuromuscular junction morphology to look for motor neuron denervation in the tibialis anterior (TA) muscles of *Matr3*^{S85C/S85C} mice. Our results reveal that *Matr3*^{S85C/S85C} mice show a significant number of motor neurons that are denervated in the muscles compared to wildtype control littermates (**Fig. 5c, d & Supplementary Fig. 9**). The neuromuscular junction defects including partially denervated motor neurons together with axonal swelling and decreased endplate size in the TA muscles in *Matr3*^{S85C/S85C} mice shown in **Fig. 5** are reminiscent of phenotypes observed in a *FUS*-ALS mouse model at early disease stage (Picchiarelli *et al. Nat Neuroscience* 2019). We acknowledge that these motor neuron phenotypes are early pathogenic events in ALS, so we have revised the manuscript accordingly. Collectively, additional evidence of motor neuron denervation provides support that *Matr3*^{S85C/S85C} mouse model is a clinically relevant model for early-stage ALS.

As suggested, we agree that it would be very important to investigate the functional consequences caused by the S85C mutation in Purkinje cells or motor neurons. Indeed, previous studies have suggested that MATR3 is involved in mRNA splicing and nuclear export. An understanding of how the S85C mutation affects MATR3's ability to regulate mRNA splicing and nuclear export would provide insights into the mechanism underlying neurodegeneration. However, as loss of MATR3 S85C immunoreactivity is observed only in specific neuronal populations, the most appropriate experiment to

address this question would be to isolate Purkinje cell or motor neuron populations. Enrichment of these selective neuronal populations prior to RNA sequencing would be required to achieve the desired number of reads to evaluate mRNA splicing changes. In addition, to determine the role of MATR3 S85C in nuclear export, we would need to know MATR3's RNA targets in these neuronal populations. While these experiments are indeed necessary and will be completed in the future, this work is beyond the scope of this manuscript.

Q3) It is unclear why the authors present data on inflammatory markers only at 30 weeks of age, which does not correlate to time points used in other studies. A more in-depth examination of the timing for inflammation in this mouse model is warranted.

A3) As suggested, we have validated whether the inflammatory genes identified in the RNA-seq data are upregulated in *Matr3^{S85C/S85C}* mice through RT-PCR using age-matched mice. Indeed, we found that inflammatory genes including *Clec7a*, *Ccl6*, *Mpeg1* and *Lpl* were significantly upregulated in the cerebellum of 8-10 week old *Matr3^{S85C/S85C}* mice compared to wildtype control littermates, and this was confirmed in both female and male animals (**Supplementary Fig. 14**).

Q4) As noted by the authors, the human MATR3 S85C disease exhibits a dominant genetic effect whereas the heterozygous knock-in mice exhibit no phenotype. This also suggests significant mechanistic differences between the knock-in rodent model and the human disease.

A4) We agree with the reviewer's comment and have added the comment regarding the existence of mechanistic differences between our knock-in mouse model and human disease in the Discussion section.

Q5) The authors often interpret their immunostaining data to indicate a loss or reduction in MATR3 expression (e.g., text describing Figure 6), but the lack of staining may not indicate a loss of MATR3 gene expression but potentially a change in protein turnover or loss of the epitope recognized by the antibody. Additional studies are needed to explore the potential change in MATR3 gene expression in this mouse model. In addition, a number of MATR3 antibodies are used throughout the study (sometimes different antibodies from figure to figure) and it's unclear if similar results are obtained by each of the antibodies.

A5) We thank the reviewer for raising this point that "a loss or reduction in MATR3 expression" in the text describing Figure 6 can be misinterpreted as a change in *Matr3* gene expression. As suggested, we have carefully revised the sentences in the text describing Figure 6.

We appreciate the reviewer's comment that there may be some confusion regarding which antibody was used for which figure. In order to present our results in a clearer manner, we have indicated directly on the figure which antibodies were used for each image (MATR3-C, ab84422; MATR3-N, HPA036565; MATR3, SC81318). We have also summarized our results into a table (**Supplementary Table 4**) to clearly show that we have obtained similar results from different antibodies.

REVIEWERS' COMMENTS:

Reviewer #1 (Remarks to the Author):

The authors have adequately responded to the points raised by the reviewers. In particular, the lack of muscle pathology, as convincingly demonstrated by the authors, is of interest. The cerebellar pathology in this mouse model is exciting and does not detract from the paper's central message. This is an attractive new ALS mouse model that draws attention to the NMJ compared to other mouse models that are more focused on different parts of the axis.

Reviewer #2 (Remarks to the Author):

The authors added a breath of new data that clearly strengthen the manuscript that is now (almost) suited for publication.

However, a few issues remain that would need to be addressed before acceptance:

- Throughout the manuscript, the authors still state that the mice develop motor neuron degeneration. This is not supported by the data presented, as their data shows a non-significant trend towards degeneration of alpha motor neurons in the lumbar spinal cord. Even in the abstract they state that the mice have motor neuron degeneration. Efforts should be made throughout the manuscript to address this issue. In particular in the discussion, they added a new paragraph that correctly address it, but didn't change the previous paragraph when they still state that "... mutant animals show degeneration of neurons that control motor function including alpha motor neurons and Purkinje cells".

- The authors present new, interesting data, on the neuromuscular junctions from TA muscle. However, from the materials, it is confusing how many mice were used for which analysis. Also, it is not clear how the authors made sure that they quantified NMJs from similar regions of the TA between the different animals.

REVIEWERS' COMMENTS

Reviewer #1 (Remarks to the Author)

The authors have adequately responded to the points raised by the reviewers. In particular, the lack of muscle pathology, as convincingly demonstrated by the authors, is of interest. The cerebellar pathology in this mouse model is exciting and does not detract from the paper's central message. This is an attractive new ALS mouse model that draws attention to the NMJ compared to other mouse models that are more focused on different parts of the axis.

A) We sincerely thank the reviewer for his/her appreciation of this new ALS mouse model and strong support for the publication of this manuscript.

Reviewer #2 (Remarks to the Author)

The authors added a breath of new data that clearly strengthen the manuscript that is now (almost) suited for publication. However, a few issues remain that would need to be addressed before acceptance:

- Throughout the manuscript, the authors still state that the mice develop motor neuron degeneration. This is not supported by the data presented, as their data shows a non-significant trend towards degeneration of alpha motor neurons in the lumbar spinal cord. Even in the abstract they state that the mice have motor neuron degeneration. Efforts should be made throughout the manuscript to address this issue. In particular in the discussion, they added a new paragraph that correctly address it, but didn't change the previous paragraph when they still state that "... mutant animals show degeneration of neurons that control motor function including alpha motor neurons and Purkinje cells".

A) As suggested, we have made the changes accordingly throughout the manuscript to describe our data clearly.

- The authors present new, interesting data, on the neuromuscular junctions from TA muscle. However, from the materials, it is confusing how many mice were used for which analysis. Also, it is not clear how the authors made sure that they quantified NMJs from similar regions of the TA between the different animals.

A) The Methods section has been revised to clearly describe the procedure for NMJ immunostaining and how quantification was performed.